# Mechanism and consequence of abnormal calcium homeostasis in Rett syndrome astrocytes

Qiping Dong[1], Qing Liu[2], Ronghui Li[1], Anxin Wang[1], Qian Bu[1], Kuan Hong Wang[2], Qiang Chang[1,3,4]*

[1]Waisman Center, University of Wisconsin-Madison, Madison, United States; [2]Unit on Neural Circuits and Adaptive Behaviors, National Institute of Mental Health, Bethesda, United States; [3]Department of Medical Genetics, University of Wisconsin-Madison, Madison, United States; [4]Department of Neurology, University of Wisconsin-Madison, Madison, United States

**Abstract** Astrocytes play an important role in Rett syndrome (RTT) disease progression. Although the non-cell-autonomous effect of RTT astrocytes on neurons was documented, cell-autonomous phenotypes and mechanisms within RTT astrocytes are not well understood. We report that spontaneous calcium activity is abnormal in RTT astrocytes in vitro, in situ, and in vivo. Such abnormal calcium activity is mediated by calcium overload in the endoplasmic reticulum caused by abnormal store operated calcium entry, which is in part dependent on elevated expression of TRPC4. Furthermore, the abnormal calcium activity leads to excessive activation of extrasynaptic NMDA receptors (eNMDARs) on neighboring neurons and increased network excitability in *Mecp2* knockout mice. Finally, both the abnormal astrocytic calcium activity and the excessive activation of eNMDARs are caused by *Mecp2* deletion in astrocytes in vivo. Our findings provide evidence that abnormal calcium homeostasis is a key cell-autonomous phenotype in RTT astrocytes, and reveal its mechanism and consequence.
DOI: https://doi.org/10.7554/eLife.33417.001

*For correspondence:
qchang@waisman.wisc.edu

**Competing interests:** The authors declare that no competing interests exist.

## Introduction

Rett syndrome (RTT) is a debilitating neurodevelopmental disorder (*Hagberg, 1985*) caused by mutations in the methyl-CpG binding protein 2 (*MECP2*) gene (*Amir et al., 1999*). RTT is a complex disease, as all key cell types (neurons, astrocytes, microglia, and oligodendrocytes) in the brain have been shown to contribute to the disease etiology (*Ballas et al., 2009*; *Chao et al., 2010*; *Derecki et al., 2012*; *Fyffe et al., 2008*; *Lioy et al., 2011*; *Luikenhuis et al., 2004*; *Maezawa and Jin, 2010*; *Maezawa et al., 2009*; *Nguyen et al., 2013*; *Samaco et al., 2009*; *Williams et al., 2014*). To fully understand the disease etiology and develop effective treatments, it is essential to define the key phenotypes, link the phenotypes with loss of MeCP2 function, and reveal the consequences of the phenotypes in each brain cell type. Moreover, it is also important to define the connection between phenotypes in different cell types.

Earlier research to understand the RTT disease etiology mostly focused on neuronal dysfunction in RTT (*Chao et al., 2010*; *Fyffe et al., 2008*; *Luikenhuis et al., 2004*; *Samaco et al., 2009*), because MeCP2 is expressed at high levels in neurons in a temporal and spatial pattern closely tracking neuronal maturation (*Akbarian et al., 2001*; *Shahbazian et al., 2002*). Not surprisingly, specific subsets of RTT-like phenotypes were observed in mouse models when *Mecp2* is deleted in specific neuronal subtypes (*Chao et al., 2010*; *Fyffe et al., 2008*; *Samaco et al., 2009*). Collectively, these studies confirm the essential role of MeCP2 in maintaining the normal function of neurons,

suggesting the loss of function or malfunction of MeCP2 in neurons as the main cause of RTT pathogenesis. Recently, a series of studies have reported that astrocytes express MeCP2, loss of MeCP2 in astrocytes causes neuronal defects in a non-cell autonomous manner, and restoring MeCP2 expression to normal level in astrocytes alone rescues some disease symptoms (*Ballas et al., 2009*; *Lioy et al., 2011*; *Maezawa et al., 2009*; *Williams et al., 2014*). Intriguingly, cell type specific deletion of *Mecp2* in astrocytes alone is not sufficient to generate most of the RTT-like phenotypes in mice (*Lioy et al., 2011*), suggesting the loss of function or malfunction of MeCP2 in astrocytes is not the initial cause of the disease, yet is required to maintain the disease progression. Although a few studies have reported gene expression changes in *Mecp2* mutant mouse astrocytes (*Forbes-Lorman et al., 2014*; *Yasui et al., 2013*), no major cell autonomous phenotype in astrocytes has been characterized in depth. Thus, identification of cell autonomous phenotypes in astrocytes not only is essential for our understanding of astrocyte dysfunction in RTT, but also will help us to determine the mechanism underlying its non-cell autonomous effects on neurons.

To identify novel cell autonomous phenotypes in RTT astrocytes, we examined intracellular calcium dynamics in astrocytes differentiated from congenic pairs of wild type and mutant RTT iPSC lines. We have used the term 'isogenic' to describe these RTT iPSC lines in our previous reports (*Ananiev et al., 2011*; *Williams et al., 2014*). However, given that the difference between wild type and mutant iPSC lines generated from the same RTT patients is the entire X chromosome, congenic is a more accurate term to define these lines. We observed abnormal spontaneous and pharmacologically evoked cytosolic calcium activities in cultured mutant RTT astrocytes, when compared to their congenic wild type controls. Similar phenotypes can be observed in *Mecp2* knockout mouse astrocytes in vitro, in situ, and in vivo. In addition, re-expression of wild type MeCP2 is sufficient to rescue the abnormal calcium activities in both the mouse and human RTT astrocytes, further confirming that these phenotypes are caused by the loss of MeCP2 function. Moreover, we report that these abnormal calcium activities are mediated by calcium overload in the endoplasmic reticulum (ER), which is caused by abnormal TRPC4-dependent store operated calcium entry (SOCE). Although we provide no direct evidence that TRPC4 is the actual SOCE channel, both loss- and gain-of-function manipulations from our study suggest TRPC4 is upstream of the SOCE and subsequent calcium phenotypes in RTT astrocytes. Finally, we show that these abnormal calcium activities in astrocytes lead to excessive activation of extrasynaptic NMDA receptors (eNMDARs) on neighboring neurons and increased network excitability; and both the abnormal astrocytic calcium activity and the excessive activation of eNMDARs are caused by the loss of *Mecp2* in astrocytes (but not neurons) in vivo. Our findings provide the first evidence that suggests abnormal calcium homeostasis as a cell autonomous phenotype in RTT astrocytes, and begin to reveal the mechanism and consequence of such a phenotype.

## Results

### Abnormal calcium activities in RTT astrocytes

Congenic pairs of wild type and mutant RTT iPSC lines carrying the common R294X mutation were differentiated into glial fibrillary acidic protein positive (GFAP+) astrocytes as described in our previous paper (*Williams et al., 2014*). On average, more than 90% of the cells were GFAP +in cultures used in our study (*Figure 1—figure supplement 1* and *Table 1*). Using the green-colored calcium-sensitive dye Fluo-4, we monitored spontaneous cytosolic calcium ($Ca^{2+}$) dynamics in live mutant (R294X-MT) and congenic wild type control (R294X-WT) astrocytes (*Videos 1–2*, *Figure 1A–B*). Ionophore A23187 was used as a loading control (*Figure 1—figure supplement 2*). A significantly higher percentage of mutant astrocytes (30 ± 3% in MT vs. 18 ± 2% in WT, n = 26 randomly selected fields

**Table 1.** Quantification of the percentage of GFAP+ cell in five differentiations from human iPSCs

| stereology # | 1 | 2 | 3 | 4 | 5 |
|---|---|---|---|---|---|
| WT | 90% | 94% | 92% | 89% | 91% |
| MT | 90% | 92% | 94% | 89% | 92% |

DOI: https://doi.org/10.7554/eLife.33417.004

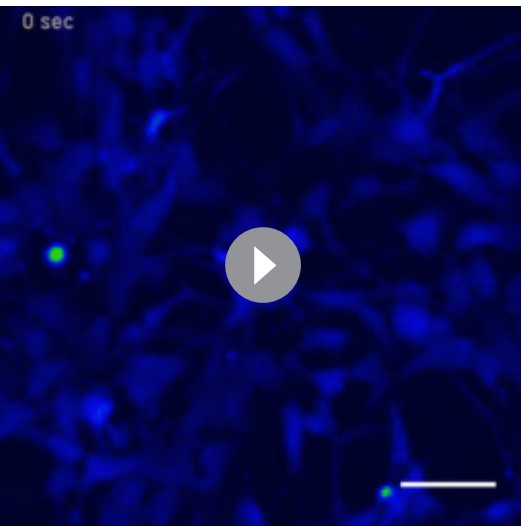

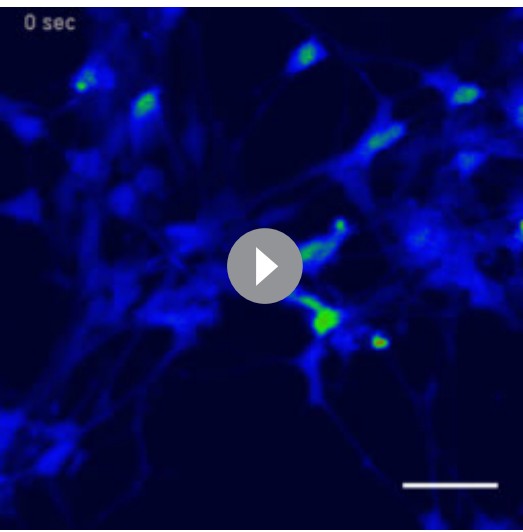

**Video 1.** Spontaneous calcium activities in wild type human astrocytes
DOI: https://doi.org/10.7554/eLife.33417.002

**Video 2.** Spontaneous calcium activities in mutant human astrocytes
DOI: https://doi.org/10.7554/eLife.33417.003

in each genotype, p=0.001) showed spontaneous oscillation in cytosolic $Ca^{2+}$ levels (*Figure 1C*). In each astrocyte showing spontaneous oscillation, we further separated and quantified the calcium transients in the soma from those in the processes (*Figure 1A*). In the soma, the mutant cells had a significantly higher frequency (0.74 ± 0.02/min in MT vs. 0.41 ± 0.01/min in WT, n = 408 cells in MT and n = 280 cells in WT, p<0.001; *Figure 1C*) and slightly higher amplitude (*Table 2*) than their congenic wild type controls. Similar changes were observed in the processes of those astrocytes (*Table 2* and *Figure 1—figure supplement 3*). Moreover, the R294X mutant astrocytes had significantly higher peak amplitude (4.60 ± 0.07 in MT vs. 4.04 ± 0.07 in WT, n = 567 cells in MT and n = 811 cells in WT, p<0.001) of ATP-induced $Ca^{2+}$ rise than the congenic wild type control (*Figure 1D–E*). The increase in ATP-induced $Ca^{2+}$ rise in human RTT mutant astrocytes was unlikely to be mediated by the P2 receptors, because no difference was detected in the expression level of any P2X or P2Y receptor family members between the wild type and mutant RTT astrocytes by RNA-seq and microarray analysis (data not shown). Similar observations of altered $Ca^{2+}$ activities were made in human astrocytes carrying the rare RTT-causing V247fs mutation (*Figure 1—figure supplement 4*). Thus, both spontaneous and pharmacologically evoked cytosolic $Ca^{2+}$ activity appeared abnormal in mutant human RTT astrocytes differentiated from patient-specific iPSCs and grown in the absence of neurons.

To determine whether the observed abnormal cytosolic $Ca^{2+}$ activities were caused by the loss of MeCP2 function, we infected mutant human RTT astrocytes with lentiviruses expressing either the green fluorescence protein (lenti-GFP) alone or GFP and wild type MECP2 together (lenti-MECP2/GFP). Three days after the infection, more than 90% of the cells were positive for GFP in both infection types (*Figure 2A*). In R294X mutant astrocytes infected with lenti-MECP2/GFP, anti-MECP2 immunoreactivity was detected in all the GFP+ cells (right, *Figure 2A*). Further quantification of MECP2 immunoreactivity in those cells revealed no difference from wild type astrocytes (*Figure 2—figure supplement 1*), suggesting the exogenous expression of MECP2 was not significant above the endogenous level. When $Ca^{2+}$ activity was recorded (*Figure 2B*), the percentage of cells with spontaneous $Ca^{2+}$ oscillation was significantly reduced by the expression of wild type MECP2, compared with expression of GFP alone, in the mutant human RTT astrocytes (20 ± 2% in MECP2/GFP vs. 36 ± 4% in GFP, n = 6 randomly selected fields in each group, p=0.006; *Figure 2C*). In the cells detected with spontaneous cytosolic $Ca^{2+}$ oscillation, both the frequency (0.27 ± 0.06/min in MECP2/GFP vs. 0.59 ± 0.06/min in GFP, n = 21 cells in MECP2/GFP and n = 32 cells in GFP, p<0.001; *Figure 2C*) and amplitude (1.65 ± 0.08 in MECP2/GFP vs. 1.75 ± 0.03 in GFP, n = 62 events in MECP2/GFP and n = 209 events in GFP, p<0.001) were significantly reduced by the exogenous

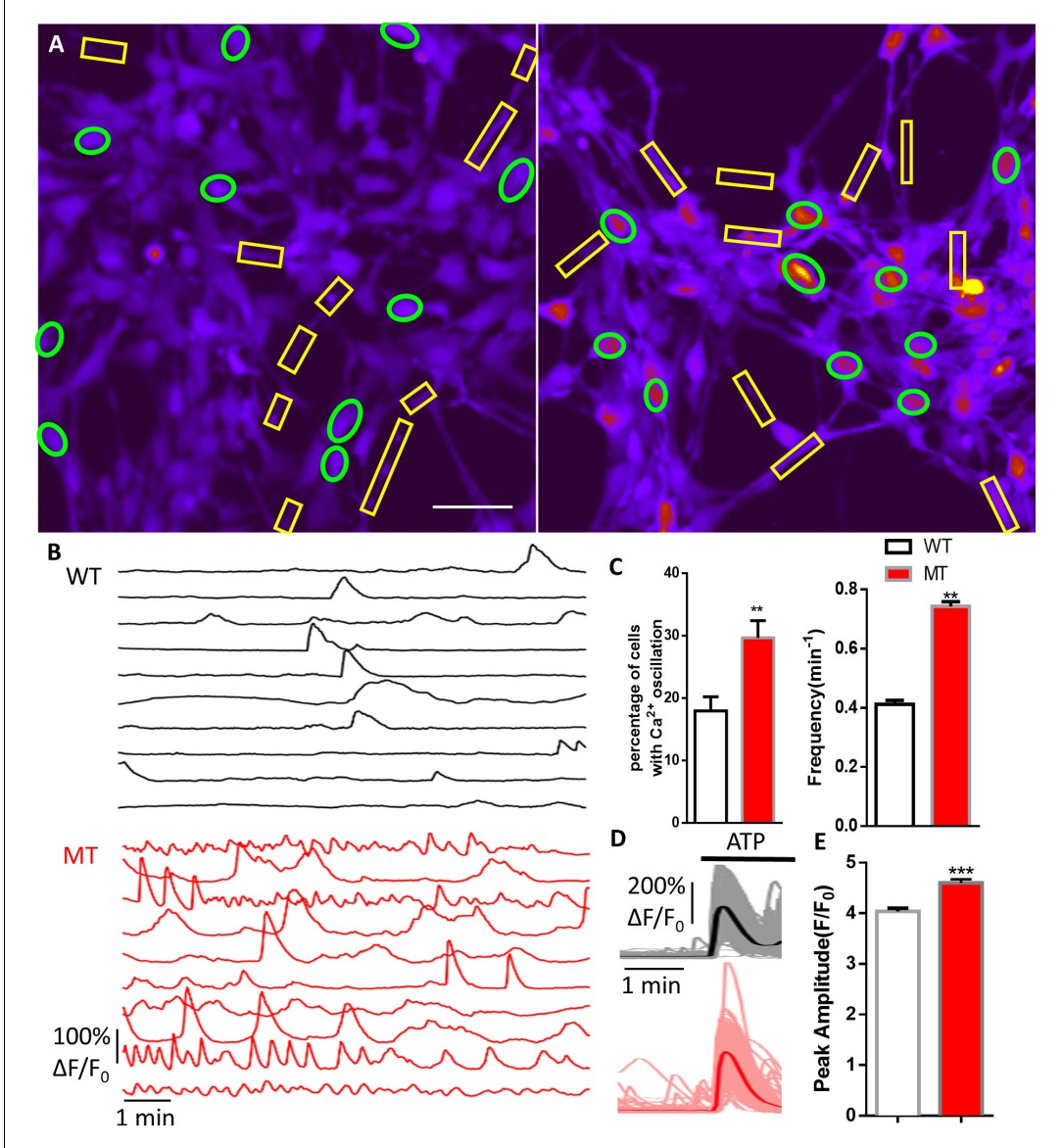

**Figure 1.** Abnormal $Ca^{2+}$ activities in mutant human RTT astrocytes. (**A**) Pseudocolored Fluo4 fluorescence images from wild type (WT, left) and MeCP2 mutant (MT, right) astrocytes differentiated from human iPSCs. Green ellipses indicate astrocyte cell soma, while yellow rectangles indicate processes. Scale bars = 50 μm. (**B**) Representative $\Delta F/F_0$ traces showing the spontaneous intracellular $Ca^{2+}$ activity from soma in (**A**). The black traces are from WT astrocytes and red traces are from MT ones. (**C**) Quantification of the percentage (left) of astrocytes showing spontaneous $Ca^{2+}$ oscillations and the frequency (right) of such oscillations. (**D**) Trace of Fluo4 fluorescence changes in wild type (WT) and mutant (MT) human astrocytes stimulated by 10 μM ATP. Average traces are shown with the solid lines. (**E**) Quantification of the peak amplitude of the ATP-evoked $Ca^{2+}$ elevations in wild type (WT) and mutant (MT) human astrocytes.

DOI: https://doi.org/10.7554/eLife.33417.006

The following source data and figure supplements are available for figure 1:

**Source data 1.** The numerical data for the graphs shown in *Figure 1C and E*.

DOI: https://doi.org/10.7554/eLife.33417.011

**Figure supplement 1.** Representative images of quantification of astrocyte differentiation efficiency from human iPSCs by co-staining with GFAP (green) and DAPI (blue) followed by stereological analysis.

DOI: https://doi.org/10.7554/eLife.33417.007

**Figure supplement 2.** Ionophore A23187 induced $Ca^{2+}$ elevations from WT and MT astrocytes.

DOI: https://doi.org/10.7554/eLife.33417.008

**Figure supplement 3.** Spontaneous $Ca^{2+}$ activity from processes of R294X wild-type (WT) and mutant (MT) human iPSC derived astrocytes.

*Figure 1 continued on next page*

*Figure 1 continued*

DOI: https://doi.org/10.7554/eLife.33417.009

**Figure supplement 4.** Abnormal spontaneous Ca$^{2+}$ activities in V247fs mutant human RTT astrocytes.
DOI: https://doi.org/10.7554/eLife.33417.010

expression of wild type MECP2 as compared with expression of GFP alone. In addition, expression of wild type MECP2 significantly reduced the elevated level of pharmacologically evoked Ca$^{2+}$ response in mutant human RTT astrocytes (3.90 ± 0.13 in MECP2/GFP vs. 4.20 ± 0.10 in GFP, n = 96 cells in MECP2/GFP and n = 123 cells in GFP, p=0.022; *Figure 2D–E*). Similar rescue effects were observed when the V247fs mutant astrocytes were used (*Figure 2—figure supplement 2*). Together, these results further support the cell autonomous nature of the abnormal Ca$^{2+}$ activities in RTT mutant astrocytes.

Since the phenotype of abnormal Ca$^{2+}$ activities have not been reported in previous studies using RTT mouse models, we next examined whether they can be observed in *Mecp2* null mouse astrocytes. Primary hippocampal astrocytes were isolated from newborn male *Mecp2* null mice (*Mecp2$^{-/y}$*) and their wild type (WT) littermates, both of which also carried the Rosa-CAG-LSL-GCaMP6s allele (http://jaxmice.jax.org/strain/024106) and the hGFAP-creERT allele (http://www.jax.org/strain/012849). The astrocytes were treated with 4-OH tamoxifen for 24 hr to induce the expression of GCaMP6s, and then imaged for GCaMP6s fluorescence as the indication of cytosolic Ca$^{2+}$ activities. Similar to our findings in human RTT astrocytes, spontaneous Ca$^{2+}$ oscillation (*Figure 3A–B*) was detected in significantly higher percentage of *Mecp2* null astrocytes than in wild type controls (67 ± 5% in *Mecp2$^{-/y}$* vs. 47 ± 6% in WT, n = 19 randomly chosen fields in both *Mecp2$^{-/y}$* and WT, p=0.01; *Figure 3C*). In cells that showed spontaneous Ca$^{2+}$ oscillation, significantly higher frequency (2.01 ± 0.04/min in *Mecp2$^{-/y}$* vs. 1.64 ± 0.03/min in WT, n = 575 cells in *Mecp2$^{-/y}$* and n = 439 cells in WT, p<0.001; *Figure 3C*) and slightly higher amplitude (*Table 2*) were observed in the soma of *Mecp2* null astrocytes than in wild type astrocytes. Since the processes were very short in primary mouse astrocyte culture, spontaneous Ca$^{2+}$ activity in processes was not quantified separately because it closely resembled that in the soma. As for pharmacologically evoked Ca$^{2+}$ response, significantly higher amplitude was detected in primary *Mecp2* null mouse astrocytes (7.25 ± 0.23 in *Mecp2$^{-/y}$* vs. 6.54 ± 0.12 in WT, n = 153 cells in *Mecp2$^{-/y}$* and n = 278 cells in WT, p=0.04) than in wild type astrocytes (*Figure 3D–E*).

Next, to explore the in situ relevance of this novel phenotype in Ca$^{2+}$ dynamics, we studied spontaneous and agonist evoked Ca$^{2+}$ activities in astrocytes in acute hippocampal slices prepared from 2 to 3 week-old *Mecp2* null mice and their wild type littermates using Fluo4. In this series of experiments, SR101 was used to label astrocytes (*Figure 4A*); TTX (1 µM) was used to suppress spontaneous neuronal activity. Similar to our observation in cultured human patient-specific iPSC derived astrocytes and primary mouse astrocytes, higher frequency (0.59 ± 0.01/min in *Mecp2$^{-/y}$* vs. 0.47 ± 0.01/min in WT, n = 337 cells in *Mecp2$^{-/y}$* and n = 293 cells in WT, p<0.001) and slightly higher amplitude (*Table 2*) of spontaneous Ca$^{2+}$ oscillation was observed in the soma of SR101+ cells in *Mecp2* null slices than in wild type slices (*Figure 4B–C*). In addition, the astrocytes in *Mecp2* null

**Table 2.** Amplitudes of spontaneous Ca$^{2+}$ elevations in different experiments

| Cell type | Wild-type | Mutant |
|---|---|---|
| R294X astrocytes | 1.660 ± 0.01624 N = 1091 | 1.808 ± 0.01882 N = 2962 |
| R294X astrocytes (processes) | 1.467 ± 0.01601 N = 2228 | 1.722 ± 0.01994 N = 2743 |
| V247fs astrocytes | 1.686 ± 0.02453 N = 902 | 1.816 ± 0.02128 N = 1760 |
| V247fs astrocytes (processes) | 1.560 ± 0.02224 N = 1825 | 1.640 ± 0.02447 N = 2312 |
| Mouse primary astrocytes | 1.787 ± 0.007193 N = 5694 | 1.822 ± 0.006209 N = 8389 |
| Astrocytes in situ | 2.010 ± 0.02998 N = 1367 | 2.151 ± 0.03079 N = 1997 |
| Astrocytes in vivo | 3.463 ± 0.1410 N = 285 | 3.045 ± 0.05410 N = 1346 |
| Astrocytes in vivo (processes) | 3.283 ± 0.03640 N = 2040 | 3.007 ± 0.01546 N = 8796 |

DOI: https://doi.org/10.7554/eLife.33417.005

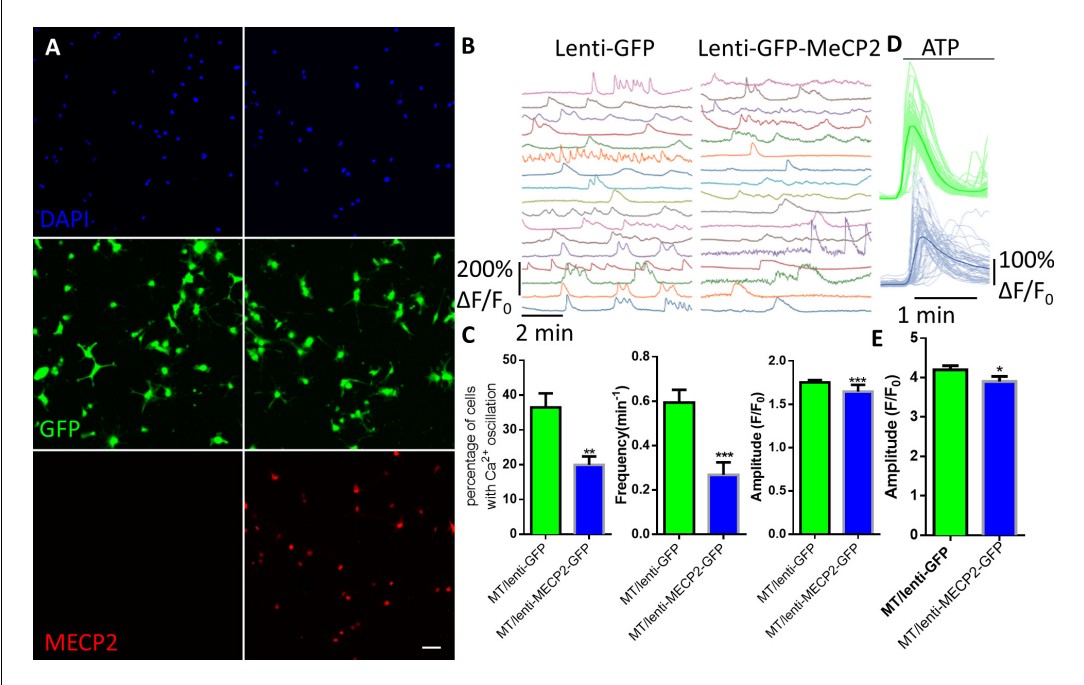

**Figure 2.** Exogenous expression of MeCP2 rescues the abnormal calcium activities in RTT astrocytes. (**A**) Representative images of mutant human RTT astrocytes infected with lentivirus expressing GFP alone (left) or with lentivirus co-expressing wild type MECP2 and GFP (right). MECP2 immunoreactivity was detected in GFAP positive cells in the right column, but not the left column. Scale bars, 50 μm. (**B**) Representative $\Delta F/F_0$ traces showing the spontaneous intracellular $Ca^{2+}$ activity in RTT mutant astrocytes infected with lentivirus expressing either GFP alone (GFP) or co-expressing GFP and wild type MECP2 together (MECP2/GFP). (**C**) Quantification of the percentage (left) of astrocytes showing spontaneous $Ca^{2+}$ oscillations and the frequency (middle) and amplitude (right) of such oscillations in RTT mutant astrocytes infected with lentivirus expressing either GFP alone (GFP) or co-expressing GFP and wild type MECP2 together (MECP2/GFP). (**D**) Trace of ATP (10 μM) evoked Fluo4 fluorescence changes in RTT mutant astrocytes infected with lentivirus expressing either GFP alone (GFP) or co-expressing GFP and wild type MECP2 together (MECP2/GFP). Average traces are shown with the solid lines. (**E**) Quantification of the peak amplitude of the ATP-evoked $Ca^{2+}$. The bar graphs in this figure show the mean ±s. e.m. *p<0.05, **p<0.01, ***p<0.001.

DOI: https://doi.org/10.7554/eLife.33417.012

The following source data and figure supplements are available for figure 2:

**Source data 1.** The numerical data for the graphs shown in *Figure 2C and E*.
DOI: https://doi.org/10.7554/eLife.33417.015

**Figure supplement 1.** Exogenous expression of MeCP2 in human iPSC derived astrocytes using lentivirus.
DOI: https://doi.org/10.7554/eLife.33417.013

**Figure supplement 2.** Exogenous expression of MeCP2 rescues the abnormal calcium activities in V247fs mutant astrocytes.
DOI: https://doi.org/10.7554/eLife.33417.014

slices had significantly higher peak amplitude (2.75 ± 0.02 in MT vs. 2.22 ± 0.09 in WT, n = 754 cells in $Mecp2^{-/y}$ and n = 238 cells in WT, p<0.001) of glutamate-induced $Ca^{2+}$ rise than astrocytes in wild type slices (*Figure 4D*).

To further rule out potential artifacts in cultured astrocytes and in acute slices, we examined spontaneous $Ca^{2+}$ activities in astrocytes of the frontal cortex in live mice carrying the GCaMP6s sensor. We mated male *Rosa-CAG-LSL-GCaMP6s* mice with either female wild type ($Mecp2^{+/+}$) or $Mecp2^{flox/flox}$ mice, injected adeno-associated virus (AAV) expressing mCherry and Cre recombinase under the human GFAP promoter (AAV8-hGFAP-mCherry-Cre) into the lateral ventricles of the postnatal day one male pups from these matings to activate the expression of GCaMP6s and delete *Mecp2* predominantly in astrocytes. Based on the mating scheme, male pups would be of the genotype of either $Mecp2^{+/y};Rosa-CAG-LSL-GCaMP6s$ (WT) or $Mecp2^{flox/y};Rosa-CAG-LSL-GCaMP6s$ (floxed). When immunostaining with MeCP2 antibody on sections prepared from 3 months old floxed mice that received AAV8-hGFAP-mCherry-Cre injection at birth, 100% of the GCaMP6s (anti-GFP) positive cells in the cortex were MeCP2 negative. In contrast, GCaMP6s positive cells from WT mice were

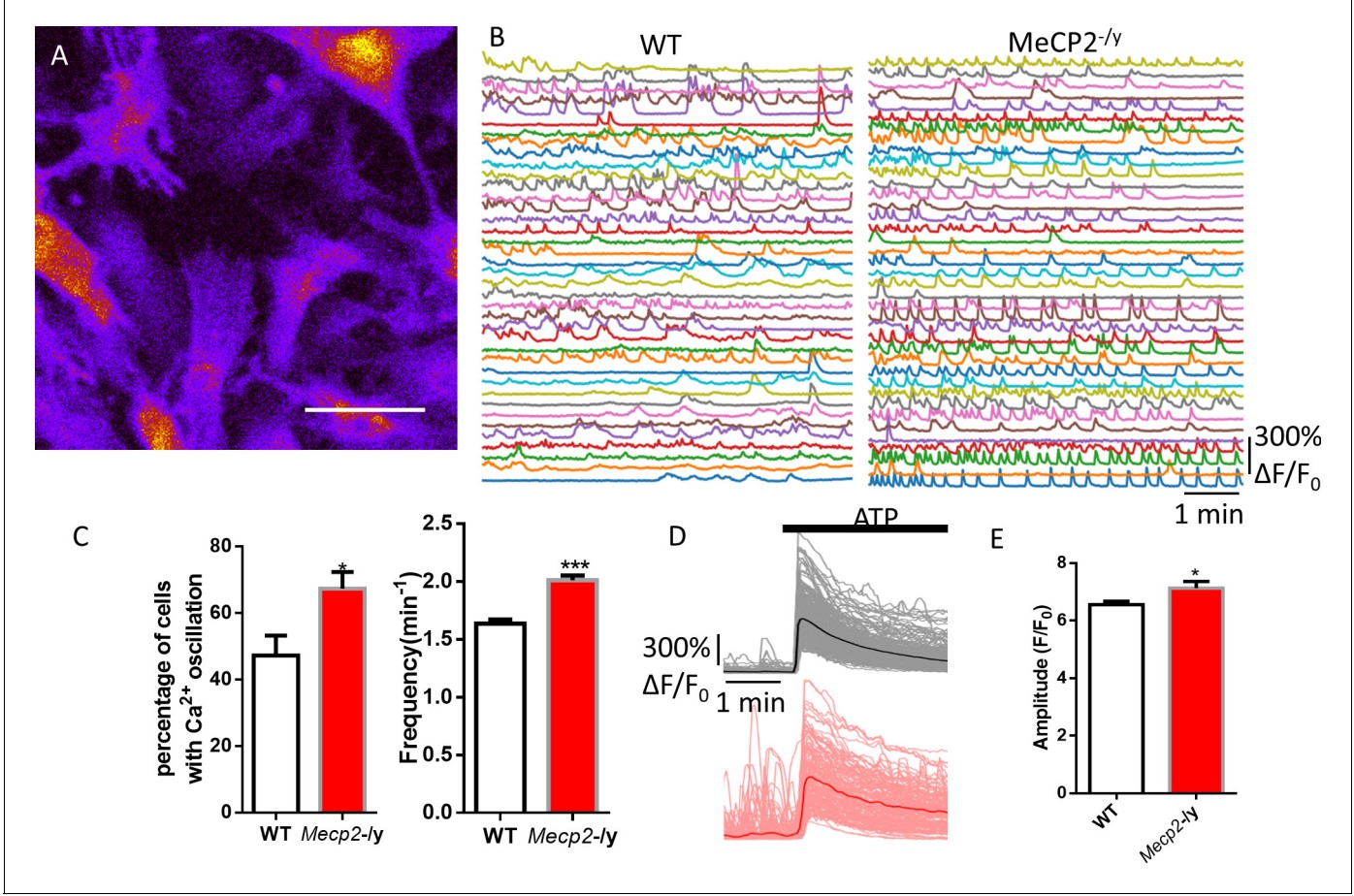

**Figure 3.** Abnormal Ca²⁺ activities in primary astrocytes isolated from *Mecp2* null mice. (**A**) The pseudocolored GCaMP6s fluorescence image from astrocytes. (**B**) Representative ΔF/F₀ traces showing the spontaneous intracellular Ca²⁺ activity in mouse primary astrocytes from wild type and *Mecp2*-/y mice. (**C**) Quantification of the percentage (left) of astrocytes showing spontaneous Ca²⁺ oscillations and the frequency (right) of such oscillations. (**D**) Trace of fluorescence changes in wild type and *Mecp2*-/y astrocytes stimulated by 10 µM ATP. Average traces are shown with the solid lines. (**E**) Quantification of the peak amplitude of the ATP-evoked Ca²⁺ elevations from wild type and *Mecp2*-/y astrocytes.
DOI: https://doi.org/10.7554/eLife.33417.016

The following source data is available for figure 3:

**Source data 1.** The numerical data for the graphs shown in *Figure 3C and E.*
DOI: https://doi.org/10.7554/eLife.33417.017

MeCP2 positive (*Figure 5—figure supplement 1*). When GCaMP6s fluorescence were recorded in mCherry-positive cells in the frontal cortex of 2–3 months-old live WT and floxed mice injected with AAV8-hGFAP-mCherry-Cre (MT) (*Video 3–4*, *Figure 5A–B*), higher frequency of spontaneous oscillation of GCaMP6s fluorescence was detected in both the soma (2.15 ± 0.11/min in MT vs. 1.44 ± 0.20/min in WT, n = 57 cells in MT and n = 18 cells in WT, p=0.003, *Figure 5C*) and processes (2.14 ± 0.04/min in MT vs. 1.69 ± 0.06/min in WT, n = 373 processes in MT and n = 110 processes in WT, p<0.001, *Figure 5—figure supplement 2*) of MT astrocytes. However, the amplitude of spontaneous astrocytic Ca²⁺ activity appeared to be slightly smaller in MT mice (*Table 2*). Taken together, the phenotype of increased frequency of spontaneous cytosolic Ca²⁺ activity was found consistently across species, as well as in vitro, in situ, and in vivo.

## Abnormal calcium load in the ER, calcium leak from the ER, baseline cytosolic calcium level, and TRPC4-dependent SOCE in RTT astrocytes

To reveal the cellular and molecular mechanisms underlying the abnormal Ca²⁺ activities in the RTT astrocytes, we first examined the source of Ca²⁺. Since the abnormal spontaneous cytosolic Ca²⁺

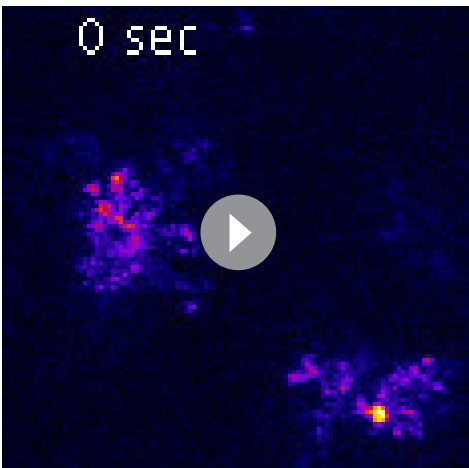

**Video 3.** Spontaneous calcium activities in astrocytes from live wild type mice
DOI: https://doi.org/10.7554/eLife.33417.020

**Video 4.** Spontaneous calcium activities in astrocytes from live *Mecp2* mutant mice
DOI: https://doi.org/10.7554/eLife.33417.021

oscillation in mutant human astrocytes persisted after $Ca^{2+}$ was removed completely from extracellular environment (0 mM $Ca^{2+}$ plus 2 mM EGTA, *Figure 6—figure supplement 1*), we measured the $Ca^{2+}$ load in the endoplasmic reticulum (ER), the major source of intracellular $Ca^{2+}$ storage. Treating astrocytes with thapsigargin (TG, 1 μM), an irreversible inhibitor of sarco-ER $Ca^{2+}$ ATPase, induced a slow elevation of $Ca^{2+}$ in the cytosol, which was mediated by the leakage of $Ca^{2+}$ from the ER. The amplitude of the TG-induced $Ca^{2+}$ elevation was significantly higher in the mutant human RTT astrocytes than in their congenic controls (125 ± 4%. in MT vs. 100 ± 3% in WT, n = 156 cells in MT and n = 178 cells in WT, p<0.001, *Figure 6A*), suggesting the mutant astrocytes had more releasable $Ca^{2+}$ in the ER. Moreover, when the kinetics of this $Ca^{2+}$ rise was analyzed, the leakage of $Ca^{2+}$ from the ER in the mutant astrocytes was much faster (time constant 54 ± 1 s in MT vs. 59 ± 1 s in WT, n = 156 cells in MT and n = 178 cells in WT, p<0.001, *Figure 6B–C*).

To accurately compare the baseline cytosolic $Ca^{2+}$ level between *Mecp2* mutant and wild type astrocytes, we prepared primary mouse astrocytes from *Mecp2$^{+/-}$* mice that carry an X chromosome-linked GFP transgene. The GFP transgene is on the same X chromosome as the wild type *Mecp2* gene. Due to random X chromosome inactivation, a mixture of GFP+ and GFP- astrocytes will be present in these cultures, with the GFP+ cells expressing the wild type MeCP2 and the GFP- cells expressing no MeCP2. When the red-colored calcium-sensitive dye Rhod-2 was used to image baseline cytosolic $Ca^{2+}$ level in the *Mecp2* null (GFP-) and wild type (GFP+) astrocytes in the same culture (*Figure 6D*), significantly higher level of Rhod-2 fluorescence was detected in the *Mecp2* null cells (154 ± 7% in GFP- vs. 100 ± 5% in GFP+, n = 30 in each group, p<0.0001; *Figure 6E*). As a control, astrocyte cultures were also prepared from *Mecp2$^{+/+}$* female mice that carry the same X chromosome-linked GFP transgene. No significant deference was detected in baseline cytosolic $Ca^{2+}$ level between the GFP+ and GFP- cells (107 ± 8% in GFP- vs. 100 ± 6% in GFP+, n = 31 in GFP- and n = 32 in GFP+, p=0.46; *Figure 6F*). These results reveal $Ca^{2+}$ overload in the ER, the faster $Ca^{2+}$ leak rate, and the elevated baseline cytosolic $Ca^{2+}$ level as significant contributors to the abnormal spontaneous and pharmacologically evoked cytosolic $Ca^{2+}$ activities in the mutant RTT astrocytes.

To further define the cellular events upstream of the abnormal $Ca^{2+}$ homeostasis, we measured $Ca^{2+}$ influx through the store operated calcium entry (SOCE) pathway, which is the primary pathway for reloading $Ca^{2+}$ into the ER in astrocytes. After depleting $Ca^{2+}$ from the ER by treating astrocytes with TG in the absence of extracellular $Ca^{2+}$ (0 mM $Ca^{2+}$ plus 2 mM EGTA), we switched bath solution to that containing 2 mM $Ca^{2+}$ and measured $Ca^{2+}$ influx. Comparing to that in the congenic control, the amplitude of $Ca^{2+}$ influx was significantly higher in the mutant human RTT astrocytes (2.93 ± 0.07 in MT vs. 2.41 ± 0.06 in WT, n = 156 cells in MT and n = 178 cells in WT, p<0.001; *Figure 6G*), suggesting an abnormal increase of SOCE in the mutant cells.

In search of the molecular mechanism underlying the increased SOCE in mutant astrocyte, we performed RNA-seq and microarray experiments to compare gene transcription profiles in the

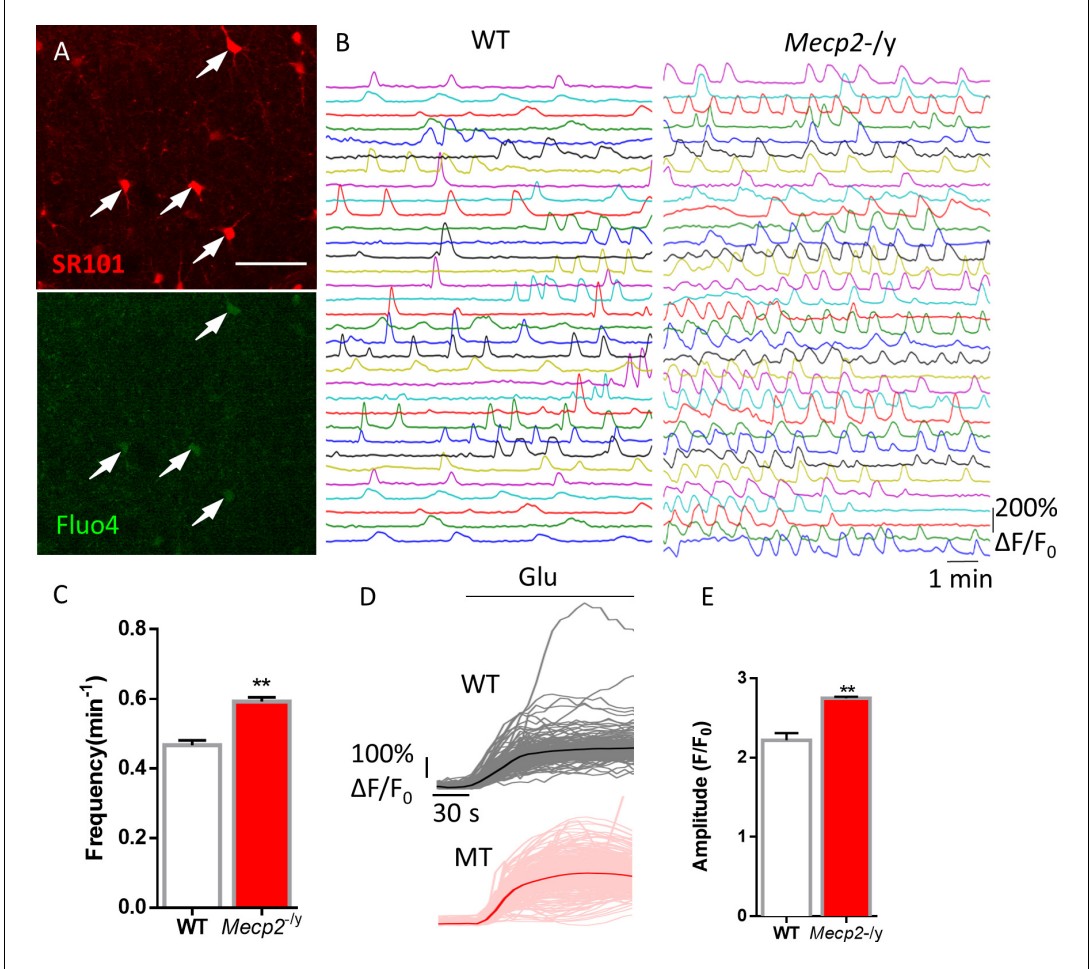

**Figure 4.** Abnormal $Ca^{2+}$ activities in astrocytes in acute brain slices prepared from *Mecp2* null mice. (A) Representative images of SR101 labeling (astrocyte marker) and Fluo4 signals in acute hippocampal slices. Note the colocalization of SR101 and Fluo4. Scale bars = 50 µm. (B) Representative $\Delta F/F_0$ traces showing the spontaneous intracellular $Ca^{2+}$ activity in SR101 positive cells in acute hippocampal slices prepared from wild type and *Mecp2$^{-/y}$* mice. (C) Quantification of the frequency of the astrocytic $Ca^{2+}$ oscillations in the slice experiments. (D) Traces of Fluo4 fluorescence changes stimulated by 100 µM glutamate in SR101 positive cells in acute hippocampal slices prepared from wild type and *Mecp2-/y* mice. Average traces are shown with solid lines. (E) Quantification of the peak amplitude of the glutamate-evoked $Ca^{2+}$ elevations. The bar graphs in this figure show the mean ±s.e.m. *p<0.05, **p<0.01, ***p<0.001.

DOI: https://doi.org/10.7554/eLife.33417.018

The following source data is available for figure 4:

**Source data 1.** The numerical data for the graphs shown in *Figure 4C and E*.

DOI: https://doi.org/10.7554/eLife.33417.019

mutant astrocytes and their congenic controls. Highly relevant to the observed SOCE phenotype, the expression of the transient receptor potential cation channel, subfamily C, member 4 (*TRPC4*), which has been previously shown to regulate SOCE in lung vascular endothelial cells (*Tiruppathi et al., 2002*), was found to be significantly higher in the mutant human RTT astrocytes (~27 fold increase by RNA-seq,~14 fold increase by microarray). Western blot analysis using protein lysates from mutant human RTT astrocytes and their congenic controls also confirmed the increased expression of TRPC4 (*Figure 6H–I*). Immunostaining with TrpC4 antibody also detected significantly higher immunoreactivity in GFAP positive astrocytes in hippocampal sections from the *Mecp2* null mouse brains as compared with that in wild type mouse brains (*Figure 6—figure supplement 2*). Chromatin immunoprecipitation analysis revealed significant MeCP2 occupancy at the promoter of *Trpc4* (*Figure 6—figure supplement 3*), suggesting direct regulation of *Trpc4* transcription by

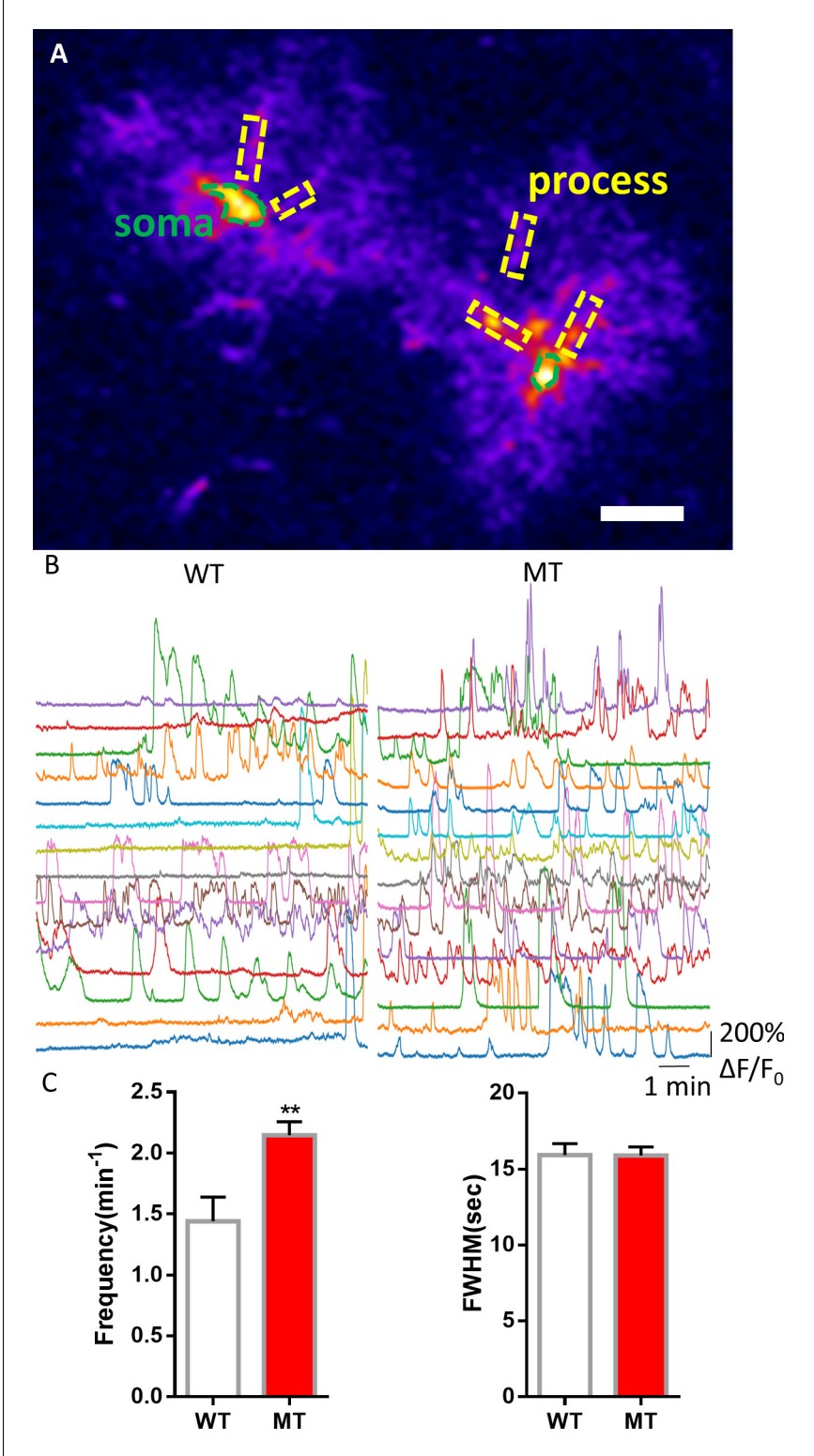

**Figure 5.** Abnormal spontaneous Ca²⁺ activities *in vivo* from *Mecp2* null astrocytes. (**A**) Representative GCaMP6s image showing two astrocytes in vivo. The somas and the processes could be clearly identified. Scale bars = 10 µm. (**B**) Representative ΔF/F₀ traces showing the spontaneous intracellular Ca²⁺ activity in the soma of astrocytes in vivo from wild type and *Mecp2* null astrocytes. (**C**) Quantification of the frequency (left) and the full width at half maximum (FWHM, right) of the astrocytic Ca²⁺ oscillations *in vivo*. The bar graphs in this figure show the mean ±s.e.m. ***p<0.001.

*Figure 5 continued on next page*

*Figure 5 continued*

DOI: https://doi.org/10.7554/eLife.33417.022

The following source data and figure supplements are available for figure 5:

**Source data 1.** The numerical data for the graphs shown in *Figure 5*.

DOI: https://doi.org/10.7554/eLife.33417.025

**Figure supplement 1.** Co-staining of MeCP2 and GCaMP6 in cortex from WT (**A**) and *MeCP2^flox/y* mice (**B**) that received AAV8-hGFAP-mCherry-Cre injection.

DOI: https://doi.org/10.7554/eLife.33417.023

**Figure supplement 2.** Quantification of spontaneous $Ca^{2+}$ activity in astrocytic processes from live WT and *Mecp2* null mice.

DOI: https://doi.org/10.7554/eLife.33417.024

MeCP2. To corroborate the expression level changes at the RNA and protein level for TRPC4, we performed whole cell patch clamp recording on the astrocytes to directly assess the current mediated by TRPC4 using a TRPC4-selective antagonist, ML204 (*Miller et al., 2011*). Significantly higher density of ML204 sensitive inward current (at −70 mV) was detected in the R294X mutant human RTT astrocytes (−0.93 ± 0.01 pA/pF in MT vs. −0.62 ± 0.08 pA/pF in WT, n = 14 cells in MT and n = 15 cells in WT, p=0.01; *Figure 6J–K*). Moreover, this current was significantly reduced when mutant human RTT astrocytes were infected with a lentivirus expressing an shRNA specific against *TRPC4* (−0.61 ± 0.10 pA/pF in MT infected with lenti-GFP-shTRPC4 vs. −1.04 ± 0.05 pA/pF in MT infected with lenti-GFP, n = 9 cells in each condition, p=0.006, *Figure 6L–M*), suggesting the current is dependent on TRPC4. The I-V curve of the ML204-sensitive current can be found in *Figure 6—figure supplement 4*. Finally, the phenotypes of increased $Ca^{2+}$ overload in the ER, faster $Ca^{2+}$ leak from the ER, increased SOCE, and increased level of TrpC4 were also observed in primary astrocytes isolated from *Mecp2* null mice (*Figure 6—figure supplement 5*).

To determine the contribution of increased TRPC4 expression to the series of $Ca^{2+}$ related phenotypes in the *Mecp2* null astrocytes, we treated these cells with ML204 for 24 hr, and then measured SOCE-mediated $Ca^{2+}$ elevation, $Ca^{2+}$ load in the ER, and recorded spontaneous cytosolic $Ca^{2+}$ activities. Comparing to untreated *Mecp2* null astrocytes, ML204-treated cells showed significantly reduced amplitude of SOCE mediated $Ca^{2+}$ elevation (3.15 ± 0.12 in ML204-treated group vs. 6.61 ± 0.86 in untreated, n = 18 in ML204-treated and n = 16 in untreated, p=0.0002; *Figure 7A*) and lower ER $Ca^{2+}$ load (1.86 ± 0.06 in ML204-treated vs. 2.47 ± 0.24 in untreated, n = 14 in ML204-treated and n = 15 in untreated, p=0.02; *Figure 7B*). More importantly, ML204 treatment significantly reduced the percentage of *Mecp2* null astrocytes showing spontaneous cytosolic $Ca^{2+}$ oscillation (23 ± 3% in ML204-treated vs. 46 ± 7% in untreated, n = 9 randomly chosen fields in ML204-treated and n = 8 randomly chosen fields in untreated, p=0.03; *Figure 7C*) and the frequency (0.20 ± 0.01/min in ML204-treated vs. 0.31 ± 0.01/min in untreated, n = 91 cells in ML204-treated and n = 203 cells in untreated, p<0.001; *Figure 7C*) and amplitude (1.64 ± 0.02 in ML204-treated vs. 1.97 ± 0.02 in untreated, n = 168 events in ML204-treated and n = 597 events in untreated, p<0.001; *Figure 7C*) of such oscillations.

In addition to pharmacological manipulation, we infected mutant RTT human astrocytes with lentivirus expressing a shRNA specific against TRPC4 to reduce its level (*Figure 7D–E*). 72 hr after the infection, we measured SOCE-mediated $Ca^{2+}$ elevation, ER $Ca^{2+}$ load, and recorded spontaneous cytosolic $Ca^{2+}$ activities. Comparing to those infected with lentivirus expressing GFP alone, the mutant astrocytes infected with lentivirus expressing shTRPC4 showed significantly reduced SOCE-mediated $Ca^{2+}$ elevations (1.72 ± 0.05 in MT/Lenti-ShTRPC4-GFP vs. 2.02 ± 0.04 in MT/Lenti-GFP, n = 23 in MT/Lenti-shTRPC4 and n = 26 in MT/Lenti-GFP, p<0.0001; *Figure 7F*), lower ER $Ca^{2+}$ load (2.37 ± 0.16 in MT/Lenti-shTRPC4-GFP vs. 2.95 ± 0.29 in MT/Lenti-GFP, n = 23 in MT/Lenti-shTRPC4 and n = 26 in MT/Lenti-GFP, p=0.04; *Figure 7G*), and lower frequency (0.42 ± 0.03/min in MT/Lenti-shTRPC4-GFP vs. 0.63 ± 0.03 in MT/Lenti-GFP, n = 86 cells in MT/Lenti-shTRPC4 and n = 79 cells in MT/Lenti-GFP, p<0.001, *Figure 7H*) of the spontaneous $Ca^{2+}$ elevations. Together, the pharmacological and molecular manipulations demonstrated that elevated TRPC4 expression is required for the abnormal calcium homeostasis in MeCP2 deficient astrocytes.

Finally, we overexpressed TRPC4 in wild type human astrocytes (*Figure 7I*), and observed increased SOCE-mediated $Ca^{2+}$ elevations (2.76. ± 0.15 in WT/GFP-TRPC4 vs. 2.36 ± 0.08 in WT/

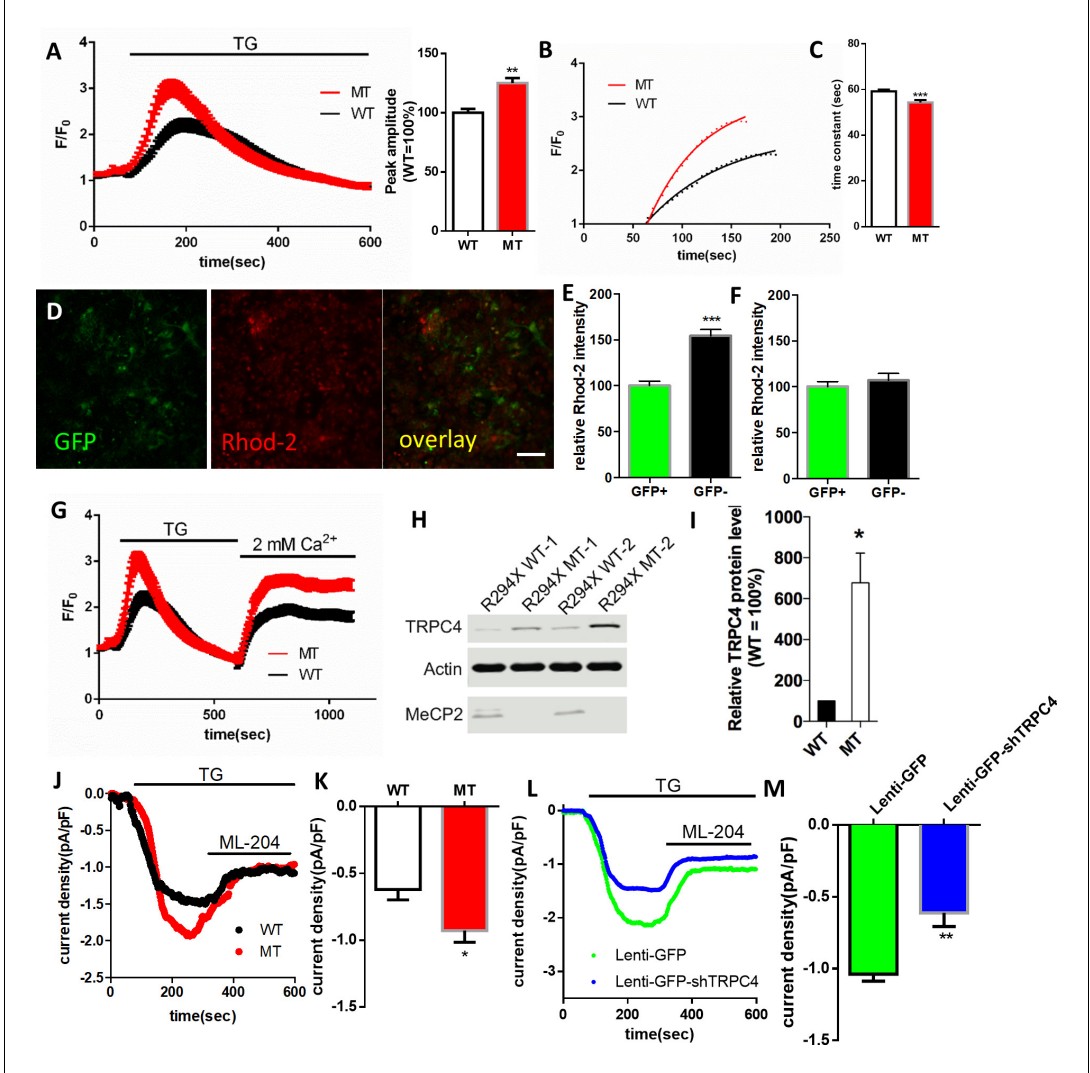

**Figure 6.** RTT astrocytes display abnormal calcium load in the ER, calcium leak from the ER, baseline cytosolic calcium level, and TRPC4-dependent SOCE. (**A**) Left, average traces of Fluo4 fluorescence changes in wild type and mutant astrocytes treated with 1 µM TG to release ER calcium. Bath solution contained 0 mM $Ca^{2+}$ plus 2 mM EGTA. Right, Quantification of the peak amplitude of TG induced $Ca^{2+}$ elevations. (**B**) The rise phase (dots) shown in (**A**) is fitted with a single exponential curve. (**C**) Quantification of the time constant of the rise phase in the left panel from congenic mutant and wild type human RTT astrocytes in response to TG. (**D**) Representative GFP and Rhod-2 images in primary astrocytes from female $Mecp2^{+/-}$ ± with a GFP transgene on the wild type X chromosome. Scale bars = 50 µm. (**E–F**) Quantification of Rhod-2 fluorescence intensity in GFP negative and GFP positive primary astrocytes isolated from either female $Mecp2^{+/-}$ ± with a GFP transgene on the wild type X chromosome (**E**) or female $Mecp2^{+/+}$ mice with a GFP transgene on the wild type X chromosome (**F**). (**G**) The average trace of Fluo4 fluorescence changes in response to extracellular $Ca^{2+}$ (2 mM) after depletion of ER $Ca^{2+}$ store using TG. (**H**) The Western blot result showing the protein level of TRPC4 in WT and MeCP2 mutant astrocytes. (**I**) Quantification of the Western blot results. (**J**) Representative traces of current density of TG-induced inward current in wild type and mutant astrocytes held at −70 mV. This current was partially blocked by TRPC4 selective antagonist, ML204. (**K**) Quantification of ML204 sensitive current from wild type and mutant astrocytes held at −70 mV. (**L**) Representative traces of current density of TG-induced inward current in mutant astrocytes infected with lentivirus expressing either GFP alone or co-expressing GFP and shTRPC4. (**M**) Quantification of ML204 sensitive current at −70 mV. The bar graphs in this figure show the mean ±s.e.m. *p<0.05, **p<0.01, ***p<0.001.

DOI: https://doi.org/10.7554/eLife.33417.026

The following source data and figure supplements are available for figure 6:

**Source data 1.** The numerical data for the graphs shown in *Figure 6A, C, E, F, I, K and M*.

DOI: https://doi.org/10.7554/eLife.33417.032

**Figure supplement 1.** Mutant human RTT astrocytes exhibit abnormal $Ca^{2+}$ activity in $Ca^{2+}$-free aCSF (0 mM $Ca^{2+}$ plus 2 mM EGTA).

DOI: https://doi.org/10.7554/eLife.33417.027

*Figure 6 continued on next page*

*Figure 6 continued*

**Figure supplement 2.** Co-staining of TRPC4 and astrocyte marker S100β on hippocampal sections from wild type (WT) and *Mecp2* knockout (*Mecp2*[-/y]) mice.

DOI: https://doi.org/10.7554/eLife.33417.028

**Figure supplement 3.** ChIP-qPCR analysis of MeCP2 occupancy on the promoter of the *TrpC4* gene in primary mouse astrocytes isolated from either *Mecp2-Flag* mice (Flag) or control mice (ctrl).

DOI: https://doi.org/10.7554/eLife.33417.029

**Figure supplement 4.** I-V curve of TG-induced ML204-sensitive current from astrocytes infected with Lenti-GFP or Lenti-GFP-shTRPC4.

DOI: https://doi.org/10.7554/eLife.33417.030

**Figure supplement 5.** *Mecp2*[-/y] astrocytes display abnormal ER $Ca^{2+}$ load, $Ca^{2+}$ leakage, store operated $Ca^{2+}$ entry, and TRPC4 expression.

DOI: https://doi.org/10.7554/eLife.33417.031

GFP, n = 52 in WT/GFP-TRPC4 and n = 55 in WT/GFP, p=0.02; *Figure 7J*), higher ER $Ca^{2+}$ load (3.65 ± 0.23 in WT/GFP-TRPC4 vs. 2.62 ± 0.13 in WT/GFP, n = 52 in WT/GFP-TRPC4 and n = 55 in WT/GFP, p=0.0001; *Figure 7K*), more cells showing spontaneous $Ca^{2+}$ oscillation (29 ± 5% in WT/GFP-TRPC4 vs. 16 ± 3% in WT/GFP, n = 13 in WT/GFP-TRPC4 and n = 12 in WT/GFP, p=0.03; *Figure 7L*), higher frequency (0.71 ± 0.04/min in WT/GFP-TRPC4 vs. 0.45 ± 0.01/min in WT/GFP, n = 74 cells in WT/GFP-TRPC4 and n = 100 cells in WT/GFP, p<0.001; *Figure 7L*) and amplitude (1.49 ± 0.04 in WT/GFP-TRPC4 vs. 1.22 ± 0.01 in WT/GFP, n = 523 events in WT/GFP-TRPC4 and n = 445 events in WT/GFP, p<0.001; *Figure 7L*) of spontaneous $Ca^{2+}$ oscillations in those cells. These results strongly suggest that elevated TRPC4 expression is sufficient to cause abnormal calcium homeostasis in wild type astrocytes.

## Abnormal calcium homeostasis in RTT astrocytes leads to excessive activation of extrasynaptic NMDA receptor activation in neighboring neurons and increased network excitability

Previous work has shown that $Ca^{2+}$ oscillation in astrocytes could lead to activation of eNMDARs (*Fellin et al., 2004*). We measured activation of eNMDAR on mouse hippocampal pyramidal neurons by whole cell patch clamp recording, and detected a significantly higher frequency of a slow inward current (SIC, *Figure 8A*, left) in acute hippocampal slices prepared from 2 to 3 weeks old *Mecp2* null mice than those from the wild type littermate control (3.88 ± 0.46/20 min in *Mecp2* null vs. 1.50 ± 0.39/20 min in WT, n = 16 in *Mecp2* null and n = 12 in WT, p=0.001; *Figure 8A*, right). The SICs were sensitive to both D-APV (selective NMDA receptor antagonist, 50 µM, *Figure 8B*, left) and ifenprodil (selective NR2B-containing NMDA receptor antagonist, 10 µM, *Figure 8B*, right), suggesting they were mediated by NR2B-containing eNMDARs. Similar increase in SIC frequency was also detected in acute hippocampal slices prepared from 6 weeks old symptomatic *Mecp2* null mice (*Figure 8—figure supplement 1*). Although similar phenotype was recently reported (*Lo et al., 2016*), the underlying cause of such increased activation of eNMDAR is not well understood.

To determine the role of abnormal $Ca^{2+}$ activities in the astrocytes in the increased frequency of SICs, membrane-impermeable BAPTA, along with a small dye that can pass through gap junctions, was injected into a single astrocyte in acute hippocampal slices prepared from the *Mecp2* null mice. Thirty minutes after injection, many astrocytes became positive for the dye in the field of injection, indicating diffusion of the dye and BAPTA into astrocytes connected through gap junctions. Whole cell patch clamp recording on hippocampal pyramidal neurons within the field of these dye-labeled astrocytes (i.e. with the intracellular chelator BAPTA) revealed a significant reduction in the frequency of SIC (2.08 ± 0.51/20 min in *Mecp2* null with BAPTA infusion into a single astrocyte vs. 5.33 ± 1.04/20 min in *Mecp2* null with no treatment, n = 12 in each group, p=0.01; *Figure 8C*), suggesting that the abnormally high calcium activity in astrocytes is required for the excessive activation of eNMDARs on neighboring neurons.

To further determine the relationship between these two phenotypes, we examined calcium activities and activation of eNMDARs in astrocyte-specific and neuron-specific *Mecp2* knockout mice. To generate cell type-specific *Mecp2* knockout mice, we mated wild type males with *Mecp2*[flox/flox] females to generate *Mecp2*[flox/y] male pups, and injected AAV viruses (AAV8-hGFAP-mCherry-Cre, AAV8-Syn-mCherry-Cre, or AAV-mCherry) into the lateral ventricles of postnatal-day-1 old *Mecp2*[flox/y] male pups. Consistent with previous report (*Kim et al., 2013*), mCherry-positive cells showed

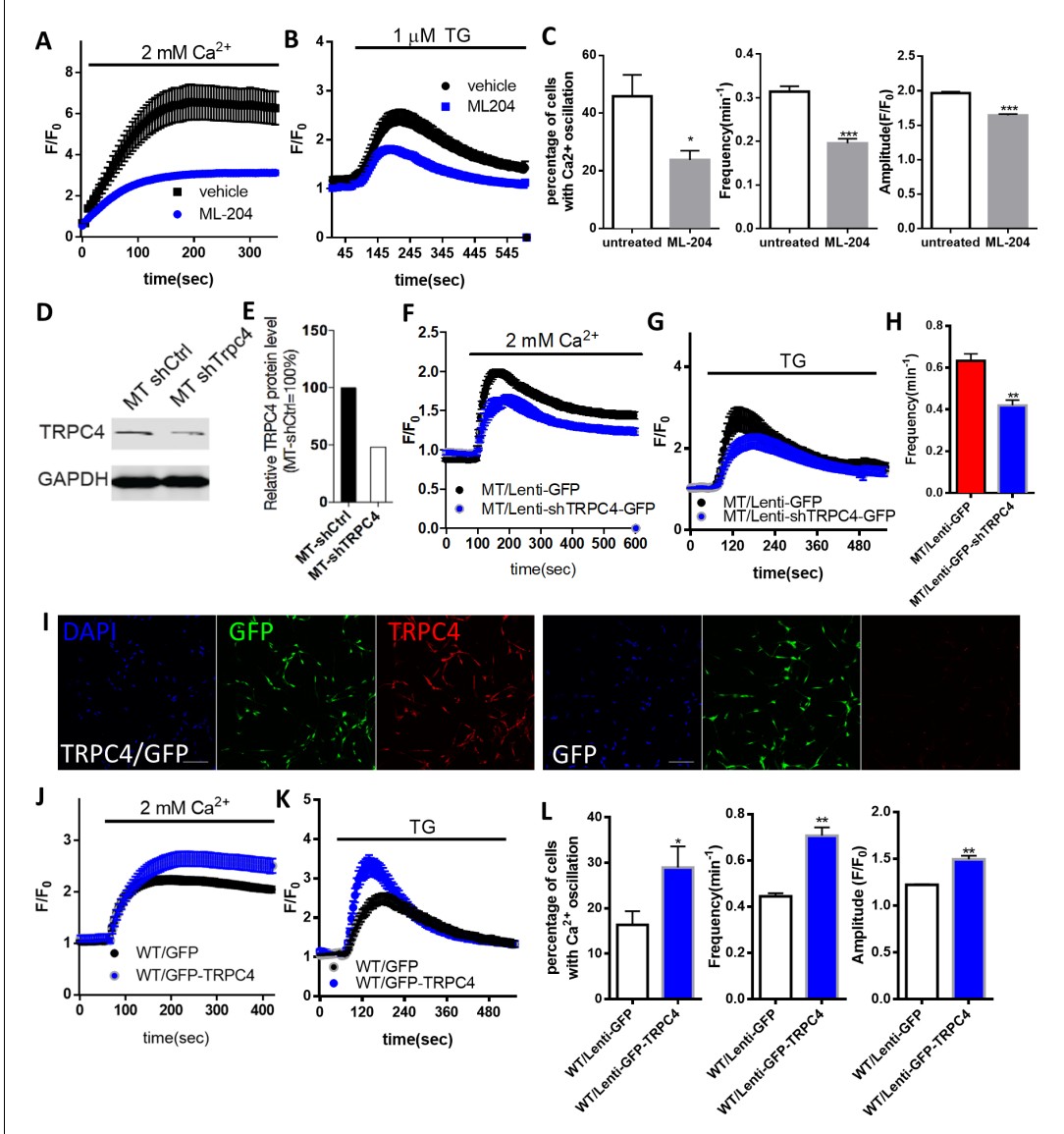

**Figure 7.** The role of TRPC4 in regulating calcium homeostasis in astrocytes. (**A**) Extracellular $Ca^{2+}$ -induced Fluo4 fluorescence changes after TG-induced ER $Ca^{2+}$ depletion in *Mecp2[-/y]* astrocytes in the absence (control) or presence of ML204. (**B**) Average traces of TG-induced Fluo4 fluorescence changes in *Mecp2[-/y]* astrocytes with or without 24 hr of ML204 treatment. (**C**) Quantification of percentage (left) of astrocytes showing spontaneous $Ca^{2+}$ oscillation and the frequency (middle) and amplitude (right) of such oscillation with or without 24 hr of ML204 treatment. (**D**) TRPC4 Western blot result showing decreased expression level of TRPC4 in mutant RTT human astrocytes after infected with lentivirus-shTrpc4-GFP, compared with astrocytes infected with lentivirus-GFP. (**E**) Quantification of the TRPC4 Western blot result. (**F**) Average traces of extracellular $Ca^{2+}$ (2 mM)-induced Rhod-2 fluorescence changes from mutant RTT astrocytes infected with lentivirus expressing either shTRPC4/GFP or GFP alone, after depletion of ER $Ca^{2+}$ store by TG pre-treatment. (**G**) Average traces of TG-induced Rhod-2 fluorescence changes from mutant RTT astrocytes infected with lentivirus expressing either shTRPC4/GFP or GFP alone. (**H**) Quantification of the frequency of spontaneous $Ca^{2+}$ elevations from mutant RTT astrocytes infected with lentivirus expressing either shTRPC4/GFP or GFP alone. (**I**) Representative images of wild type human RTT astrocytes infected with lentivirus co-expressing TRPC4 and GFP (left) or with lentivirus expressing GFP alone (right). Note anti-TRPC4 immunoreactivity is higher in astrocytes infected with lenti-GFP/TRPC4 than in those infected with lenti-GFP. Scale bar = 50 μm. (**J**) Average traces of extracellular $Ca^{2+}$ (2 mM)-induced Rhod-2 fluorescence changes from wild type astrocytes infected with lentivirus expressing either GFP/TRPC4 or GFP alone, after depletion of ER $Ca^{2+}$ store by TG pre-treatment. (**K**) Average traces of TG-induced Rhod-2 fluorescence changes from wild type astrocytes infected with lentivirus expressing either GFP/TRPC4 or GFP alone. (**L**) Quantification of percentage (left) of astrocytes infected with lentivirus expressing either GFP/TRPC4 or GFP alone that showed spontaneous $Ca^{2+}$ oscillation and the frequency (middle) and amplitude (right) of such oscillations. The bar graphs in this figure show the mean ±s.e.m. *p<0.05, **p<0.01, ***p<0.001.

DOI: https://doi.org/10.7554/eLife.33417.033

*Figure 7 continued on next page*

Figure 7 continued

The following source data is available for figure 7:

**Source data 1.** The numerical data for the graphs shown in *Figure 7C, H and L*.

DOI: https://doi.org/10.7554/eLife.33417.034

widespread presence in the forebrain (including cortex and hippocampus), suggesting robust distribution of AAV in this method. Immunohistological analysis of sections from *Mecp2$^{flox/y}$* male mice injected with AAV8-hGFAP-mCherry-Cre showed that 100% of mCherry-positive cells were positive for GFAP (right, *Figure 8D*, *Table 3*) and negative for MeCP2 (*Figure 8—figure supplement 2*), and that ~ 27% of GFAP-positive cells were positive for mCherry (*Table 3*), suggesting successful deletion of *Mecp2* in about one third of the astrocytes in the forebrain. Similar analysis of sections from *Mecp2$^{flox/y}$* male mice injected with AAV8-Syn-mCherry-Cre showed that 100% of mCherry-positive cells were positive for NeuN (left, *Figure 8D*, *Table 3*) and negative for MeCP2 (*Figure 8—figure supplement 2*), and that ~ 74% of NeuN-positive cells were positive for mCherry (*Table 3*), suggesting successful deletion of *Mecp2* in about more than two thirds of the neurons in the forebrain. Further double staining with either anti-GFAP and anti-MeCP2 antibodies or anti-NeuN and anti-MeCP2 antibodies on those brain sections (*Figure 8—figure supplement 2*) revealed that, while almost all mCherry-positive cells were indeed negative for MeCP2 in mice receiving either AAV virus, there was a negligible number (2%) of NeuN-positive cells that were negative for MeCP2 in the hippocampus of the AAV-GFAP-mCherry-Cre injected mice, and a small number (7%) of GFAP-positive cells that were negative for MeCP2 in the hippocampus of the AAV-hSyn-mCherry-Cre injected mice (*Table 4*). Thus, mice with predominantly astrocyte-specific deletion of *Mecp2* and predominantly neuron-specific deletion of *Mecp2* were generated. *Mecp2$^{flox/y}$* male mice injected with AAV-mCherry (no Cre recombinase) were used as wild type control for this series of experiments.

Abnormal spontaneous calcium activity in astrocytes (*Figure 8E*, n = 34 in AAV-mCherry group, n = 36 in AAV-Syn-mCherry-Cre group, and n = 47 in AAV-hGFAP-mCherry-Cre group; ANOVA: F = 14.11, p<0.0001) and excessive activation of extrasynaptic NMDA receptors in neighboring neurons (*Figure 8F*, n = 38 in AAV-mCherry group, n = 21 in AAV-Syn-mCherry-Cre group, and n = 30 in AAV-hGFAP-mCherry-Cre group; ANOVA: F = 4.23, p<0.018) were only observed in predominantly astrocyte-specific, but not in predominantly neuron-specific, *Mecp2* knockout hippocampal slices. Results from this series of genetic manipulations strongly suggest that the abnormal calcium activities in astrocytes are caused by the loss of MeCP2 function in astrocytes, and can lead to excessive activation of extrasynaptic NMDA receptors on neighboring neurons (even when these neighboring neurons are wild type).

Beyond eNMDAR activation, several previous studies have suggested that increased astrocytic Ca$^{2+}$ activity may lead to a hyperexcitable network (*Gómez-Gonzalo et al., 2010*; *Kuchibhotla et al., 2009*; *Tian et al., 2005*; *Wetherington et al., 2008*). Such a hypothesis is highly relevant to RTT, because 80–90% of RTT patients have seizures (*Jian et al., 2007*), and spontaneous seizures have also been reported in RTT mice (*D'Cruz et al., 2010*). As the first step to explore a role for abnormal astrocytic Ca$^{2+}$ activity in leading to seizure in RTT, we adopted a widely used slice model of seizure (or epileptiform activity or network hyper-excitability) by adding the GABA$_A$ receptor antagonist biccucullin and increasing bath Ca$^{2+}$ concentration (*Borck and Jefferys, 1999*; *McLeod et al., 2013*; *Ratté et al., 2011*; *Tian et al., 2005*).

In this model, whole cell patch clamp recording in CA1 pyramidal neurons readily detected epileptiform activity in slices from 5 to 8 week-old wild type and *Mecp2$^{-/y}$* mice (*Figure 8G*). Comparing with wild type, epileptiform activity in *Mecp2$^{-/y}$* neurons had shorter latency (4.5 ± 0.4 min in *Mecp2$^{-/y}$* vs. 6.5 ± 0.6 min in wild type, *Figure 8H*), higher frequency (5.6 ± 0.4 Hz in *Mecp2$^{-/y}$* vs. 3.5 ± 0.5 Hz in wild type, *Figure 8H*), longer duration (755 ± 30 ms in *Mecp2$^{-/y}$* vs. 621 ± 39 ms in wild type, *Figure 8H*), and higher amplitude (364 ± 15 pA in *Mecp2$^{-/y}$* vs. 279 ± 28 pA in wild type. *Figure 8H*). More importantly, intracellular infusion of membrane impermeable BAPTA into a single astrocyte significantly reduced epileptiform activity in neighboring neurons (latency: 4.5 ± 0.4 min in *Mecp2$^{-/y}$* vs. 5.7 ± 0.4 min in *Mecp2$^{-/y}$* with BAPTA infusion; frequency: 5.6 ± 0.4 Hz in *Mecp2$^{-/y}$* vs. 3.8 ± 0.5 Hz in *Mecp2$^{-/y}$* with BAPTA infusion; duration: 755 ± 30 ms in *Mecp2$^{-/y}$* vs. 633 ± 31 ms in *Mecp2$^{-/y}$* with BAPTA infusion. *Figure 8H*). 7–10 neurons from 3 to 4 mice were analyzed from each

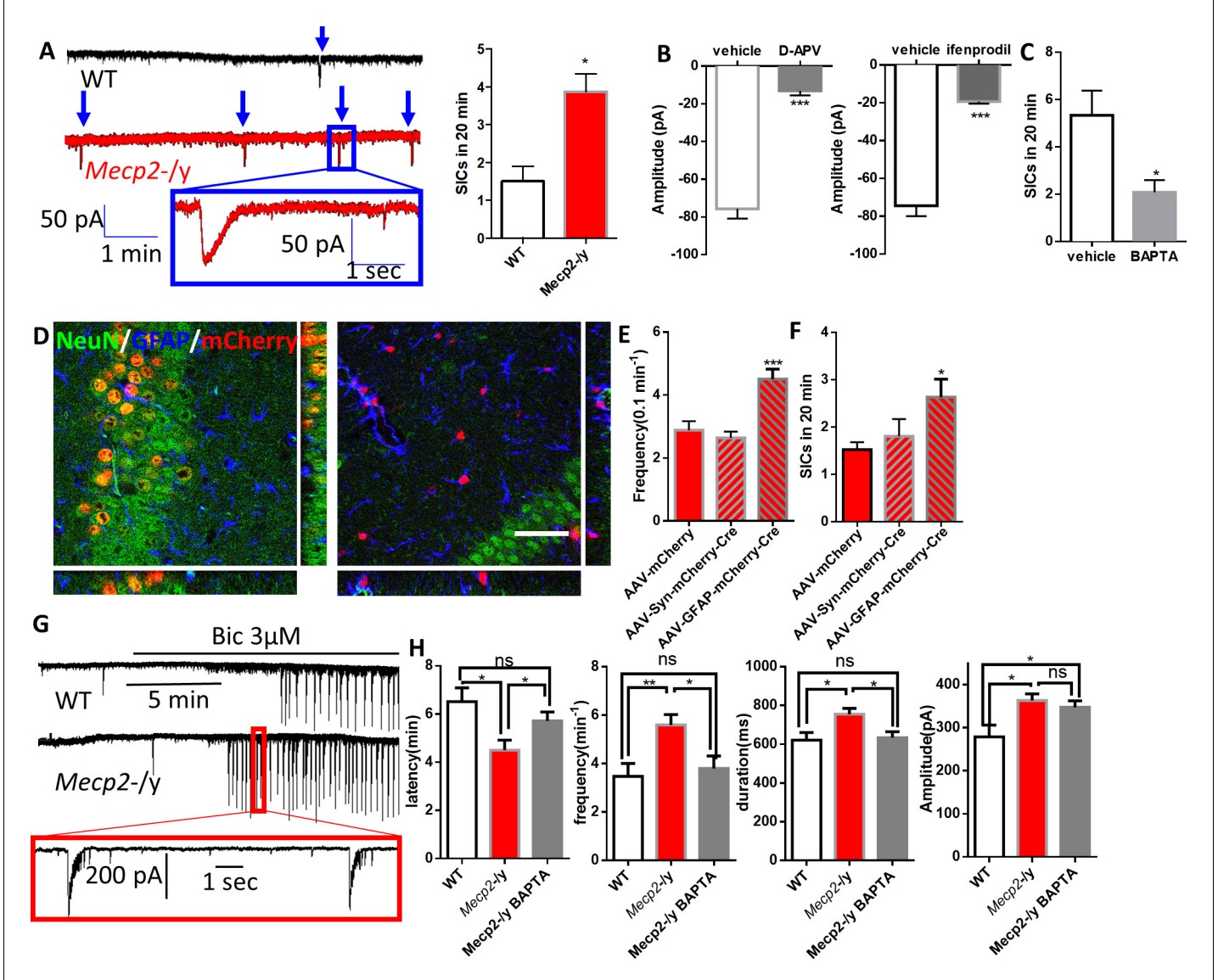

**Figure 8.** Abnormal Ca²⁺ activities in astrocytes lead to excessive activation of extrasynaptic NMDA receptors in neighboring neurons and increased network excitability. (**A**) Left: representative traces of whole-cell patch clamp recording from CA1 pyramidal neurons in acute hippocampal slices prepared from wild type and *Mecp2⁻/y* mice showing slow inward currents (SICs). Arrows indicate all SICs. Inset: magnified view of the recording. Right: quantification of the SIC frequency in neurons from wild type and *Mecp2⁻/y* mice. (**B**) Quantification of the amplitude of SICs of neurons under control condition and in the presence of D-APV (left), an antagonist of NMDA receptors, or ifenprodil (right), a selective antagonist of NR2B-containing NMDA receptors. (**C**) Quantification of the SIC frequency in neurons adjacent to astrocytes with or without intracellular infusion of BAPTA. (**D**) Orthogonal (x-y view, y-z view, and x-z view) projections of z-scanning images showing immunofluorescence of NeuN and GFAP in regions around the hippocampus. The left image is from a *Mecp2ᶠˡᵒˣ/y* mouse injected with AAV-hSyn-mCherry-Cre, while the right image is from a *Mecp2ᶠˡᵒˣ/y* mouse injected with AAV-GFAP-mCherry-Cre. Note that all of the mCherry positive cells are NeuN positive in the left image, while all of the mCherry positive cells are GFAP positive in the right image. Scale bars = 50 μm. (**E**) Quantification of the frequency of the spontaneous Ca²⁺ oscillations in astrocytes from *Mecp2ᶠˡᵒˣ/y* mice injected with AAV-mCherry, AAV-hSyn-mCherry-Cre, or AAV-GFAP-mCherry-Cre. ***p<0.001 vs. AAV-mCherry. (**F**) Quantification of the SIC frequency from *Mecp2ᶠˡᵒˣ/y* mice injected with AAV-mCherry, AAV-hSyn-mCherry-Cre, or AAV-GFAP-mCherry-Cre. *p<0.05 vs. AAV-mCherry. (**G**) Representative patch clamp recordings in CA1 pyramidal neurons showing characteristic spontaneous epileptiform bursting activity in response to application of the GABA_A-receptor antagonist bicuculline (3 μM) from wild-type (top) and Mecp2⁻/y (bottom) mice. (**H**) Quantification of the latency, frequency, duration and the amplitude of the epileptiform activity. The latency is defined as the time elapsed between Bic application and epileptiform activity onset. The bar graphs in this figure show the mean ±s.e.m. *p<0.05, **p<0.01, ***p<0.001.

DOI: https://doi.org/10.7554/eLife.33417.035

The following source data and figure supplements are available for figure 8:

**Source data 1.** The numerical data for the graphs shown in *Figure 8A, B, C, E, F and H*.

*Figure 8 continued on next page*

*Figure 8 continued*

DOI: https://doi.org/10.7554/eLife.33417.038

**Figure supplement 1.** Excessive activation of extrasynaptic NMDA receptors in acute hippocampal slices from *Mecp2* knockout (*Mecp2⁻/y*) mice.

DOI: https://doi.org/10.7554/eLife.33417.036

**Figure supplement 2.** Immunohistological analysis of AAV-injected mice.

DOI: https://doi.org/10.7554/eLife.33417.037

genotype/treatment in this series of experiments. Similar results were obtained recording field potentials under the same conditions described in *Figure 8G–H* (data not shown). Together, these data reveal a potential link between increased astrocytic $Ca^{2+}$ activity and increased network excitability. Since such increased network excitability may underlie seizure-a specific RTT symptom, these data further highlight the functional significance of abnormal $Ca^{2+}$ activity in RTT.

## Discussion

Since the initial report that astrocytes express MeCP2 and that *Mecp2* null astrocytes have non-cell autonomous influence on wild type neurons (*Ballas et al., 2009*), many gene expression changes (*Maezawa et al., 2009*; *Okabe et al., 2012*; *Yasui et al., 2013*) and a few cellular phenotypes have been identified in *Mecp2* null astrocytes (*Maezawa et al., 2009*; *Nectoux et al., 2012*). However, it is not clear how these alterations contribute to RTT pathogenesis either by directly changing astrocyte functions or by indirectly changing neuronal functions. In addition, no major cell autonomous phenotype in astrocytes has been clearly defined, with its underlying mechanism and downstream consequences examined in detail.

A cell autonomous phenotype in astrocytes is inherently difficult to define in vivo, because astrocytes normally develop in the presence of neurons and intimately interact with neurons in the intact brain. To circumvent this issue, we first turned to human iPSC-differentiated astrocytes because they are differentiated and maintained in the absence of neurons. Using astrocytes differentiated from congenic pairs of human RTT patient specific iPSCs, we discovered that both the spontaneous and the pharmacologically evoked cytosolic calcium activities are abnormal in mutant RTT astrocytes. We then confirmed that similar phenotypes could be found in *Mecp2* null mouse astrocytes in vitro, in situ, and in vivo. While significant difference exists across the astrocyte populations included in our study (species [human and mouse], developmental stage [embryonic and postnatal], and system [in vitro, in situ, and in vivo]), the core phenotype of abnormal astrocytic calcium activity remains consistent. As for the underlying mechanism, we provide evidence that the abnormal calcium activity is mediated by calcium overload in the endoplasmic reticulum (ER) caused by TRPC4-dependent abnormal store operated calcium entry (SOCE). Although our findings don't directly reveal whether TRPC4 is a component of the SOCE channel, they do suggest elevated TRPC4 expression in mutant RTT astrocytes is upstream of the SOCE, ER, and cytosolic calcium phenotypes. As for the downstream consequences, we demonstrate that these abnormal calcium activities lead to excessive activation of the eNMDARs on neighboring neurons and increased network excitability. Together, our results not only identify a novel cell autonomous phenotype and its underlying mechanism in RTT astrocytes, but also reveal a functional link between the astrocyte phenotype and a novel neuronal phenotype in RTT models and network hyper-excitability that may underlie a significant RTT symptom. For future studies, we hypothesize a linear model in the following order: first, loss of MeCP2 expression in astrocytes leads to increased expression of TRPC4 in astrocytes; second, increased expression of TRPC4 in astrocytes leads to calcium overload in the ER of astrocytes; third, calcium overload in the ER of astrocytes leads to increased cytosolic calcium activity in the astrocytes; fourth, increased cytosolic calcium activity in astrocytes leads to excessive activation of eNMDARs in neighboring neurons and increased network excitability (summarized in *Figure 9*).

The abnormal calcium homeostasis in mutant RTT astrocytes manifests in multiple interconnected cellular events, including elevated SOCE, $Ca^{2+}$ overload in the ER, elevated baseline cytosolic $Ca^{2+}$ level, and abnormal spontaneous and evoked $Ca^{2+}$ activities. Both molecular and pharmacological inhibition of the TRPC4 channel in mutant RTT astrocytes rescue the phenotypes of increased SOCE, ER $Ca^{2+}$ overload and abnormal cytosolic $Ca^{2+}$ activities. Conversely, overexpression of TRPC4 in wild type astrocytes can cause the mutant phenotypes. Together, these results strongly suggest that

**Table 3.** Histological results of mice injected with AAVs (I)

| | Mice injected with AAV-GFAP-mCherry-Cre | Mice injected with AAV-hSyn-mCherry-Cre |
|---|---|---|
| %GFAP + cells in all mCherry + cells | 100 | 0 |
| %NeuN + cells in all mCherry + cells | 0 | 100 |
| %mCherry + cells in all GFAP + cells | 28 ± 3 | 0 |
| %mCherry + cells in all NeuN + cells | 0 | 74 ± 10 |

DOI: https://doi.org/10.7554/eLife.33417.039

increased TRPC4 expression is upstream of the altered SOCE, and the subsequent ER and cytosol $Ca^{2+}$ phenotypes. As most of our TRPC4-related data came from in vitro experiments, future in vivo studies are needed to ascertain the functional significance of increased TRPC4 expression in RTT disease progression. In addition, it is worth noting that other TRP channels such as TRPA1, TRPC, TRPV4 have been implicated in modulating $Ca^{2+}$ dynamics in astrocytes (*Ma et al., 2016*; *Molnár et al., 2016*; *Shibasaki et al., 2014*; *Shigetomi et al., 2013*; *Shigetomi et al., 2011*). Yet, their expression remained unchanged in mutant human RTT astrocytes (data not shown). Of course, other deregulated genes and pathways due to the loss of MeCP2 function may also contribute to these phenotypes, providing interesting topics for future studies to further illustrate the central role of abnormal calcium homeostasis in astrocyte dysfunction in RTT. Furthermore, it is worth noting that abnormal calcium activities in astrocytes have been implicated in other neurological diseases, such as Alzheimer's disease(*Kuchibhotla et al., 2009*), suggesting abnormal calcium homeostasis in astrocytes may be a common pathological event in neurological diseases. However, the molecular mechanisms underlying the abnormal calcium activities may be different in different diseases. While the P2Y1 receptor is shown to mediate the astrocytic hyperactivity in Alzheimer's disease (*Delekate et al., 2014*), we have identified TRPC4 as a major contributor to the abnormal calcium homeostasis in RTT astrocytes. Since $Ca^{2+}$ is a pleiotropic signal for a diverse range of cellular functions, abnormal $Ca^{2+}$ homeostasis can lead to additional functional consequences in mutant RTT astrocytes. Future studies are therefore needed to define the full spectrum of cellular deficits in RTT astrocytes that are downstream of the altered $Ca^{2+}$ signaling.

Up to now, a major challenge in understanding the contribution of astrocytes to RTT pathogenesis has been the missing link between astrocyte dysfunction and neuronal dysfunction and RTT symptoms. We showed that intracellular infusion of a membrane-impermeable calcium chelator, which abolishes calcium activities in astrocytes only, reduced the excessive activation of eNMDARs on neighboring neurons and rescued network hyper-excitability. Moreover, we provided genetic evidence that the loss of MeCP2 in astrocytes is both necessary and sufficient to cause the abnormal calcium activity in astrocytes and the excessive activation of eNMDARs in neighboring neurons. These results strongly suggest that this novel neuronal phenotype is dependent on a cell autonomous phenotype in the astrocytes. Thus, our findings provide a direct functional link between astrocyte dysfunction and neuronal dysfunction in RTT.

Although it remains debatable whether excessive activation of eNMDARs leads to increased neuronal synchrony and seizure in general (*Angulo et al., 2004*; *Fellin et al., 2004*; *Tian et al., 2005*), increased synchronous neuronal firing has recently been observed in the hippocampus of both male $Mecp2^{-/y}$ and female $Mecp2^{+/-}$ mice(*Lu et al., 2016*). Thus, the relationship between the abnormal astrocytic calcium activity, the excessive activation of eNMDARs, and increased neuronal synchrony

**Table 4.** Histological results of mice injected with AAVs (II)

| | Mice injected with AAV-GFAP-mCherry-Cre | Mice injected with AAV-hSyn-mCherry-Cre |
|---|---|---|
| %MeCP2- cells in all mCherry + cells | 98 ± 0.8 | 97 ± 1.5 |
| %MeCP2- cells in all GFAP + cells | 35 ± 3 | 7 ± 1.7 |
| %MeCP2- cells in all NeuN + cells | 2 ± 0.1 | 75 ± 11 |

DOI: https://doi.org/10.7554/eLife.33417.040

is worth further studying in the context of RTT. While 80% of RTT patients have seizures (*Jian et al., 2007*), the cellular and molecular mechanisms underlying this symptom are not well understood. Even if the excessive activation of eNMDARs and/or the increased neuronal synchrony may not be directly linked to the seizure phenotype in RTT, our data linking the abnormal calcium activities in RTT astrocytes to increased network excitability still provide a new avenue for investigating this often difficult-to-manage symptom in RTT. At this moment, the exact molecular events linking abnormal calcium activities in astrocytes and altered neuronal and network properties remain elusive. One obvious candidate is glutamate, because previous studies have detected elevated glutamate level in primary cultures of astrocytes and microglia prepared from RTT mice (*Maezawa and Jin, 2010*; *Okabe et al., 2012*). Beyond glutamate, astrocytic calcium is known to modulate inhibitory synaptic efficacy by controlling GABA level through astrocytic GAT-3 (*Shigetomi et al., 2011*), and to modulate basal synaptic transmission through the release of purines (*Panatier et al., 2011*). Both mechanisms may help explain how abnormal calcium activities in RTT astrocytes cause increased network excitability. Therefore, more systematic approaches are needed in future studies to fully characterize the changes in factors present in RTT astrocyte conditioned medium (due to alteration in either release or uptake by astrocytes) and reveal how those changes underlie the non-cell autonomous influence on neurons.

## Materials and methods

### Cell lines

This study used congenic pairs of wild type and mutant Rett syndrome patient specific induced pluripotent stem cell lines carrying the R294X and the V247fs mutations. The cell lines were routinely tested to ensure that the cell lines were of human origin, were not contaminated with other cell lines, and were not contaminated with other microbial species including bacteria, fungi, or mycoplasma. Human identity testing was performed using Short Tandem Repeat (STR) testing. This test method is capable of providing a unique genetic fingerprint that is traceable back to a single human donor. In addition, STR is capable of detecting other contaminating cell types, either other human cell lines or other species, at a very low level. Isoenzyme analysis could also be performed to verify species of origin for cell lines. Testing was also routinely performed to test for microbial and fungal contamination using test methods that are typically used for testing clinical-grade cell lines. In addition, testing for mycoplasma contamination was be performed using either MycoAlert (Lonza), qPCR testing, or direct/indirect culture methods. These test methods provided a very high level of assurance that cell lines were not contaminated with other mammalian, bacterial, fungal, or mycoplasma cell lines.

In addition, for congenic pairs of iPSC lines derived from the same human RTT patient, the X chromosome inactivation status and the allelic expression of MECP2 were closely monitored to ensure correct genotype identification. These methods have been described in detail in our previous studies (*Ananiev et al., 2011*; *Williams et al., 2014*).

### Human astroglial differentiation

Human astrocytes were differentiated from astroglial progenitors as previously described (*Williams et al., 2014*). Briefly, Terminal differentiation into astrocytes from astroglial progenitors was achieved by dissociating progenitor spheres into single cells with Accutase, and plating on coverslips pre-coated Poly-Lysine at a density of $2 \times 10^5$ cells per ml in astrocytes differentiation media: DMEM/F12 with 1% N2, $1 \times$ NEAA, $1 \times$ pen/strep and 2 µg/ml heparin (Sigma H3149), and CNTF (BioSensis). To facilitate cells attachment, 10% FBS (Gibco) was added. The next day, the media was replaced by fresh ADM without FBS. Differentiated progenitors were then fed every other day and used for experimentation 7 days after the start of terminal differentiation. To improve cell attachment, in some experiments, dissociated astroglial progenitor cells were plated on Matrigel-coated plates. The medium used for plating was DMEM/F12 with 1% N2, $1 \times$ NEAA, $1 \times$ pen/strep and 2 µg/ml heparin, supplemented with 10 ng/ml BMP4, LIF, CNTF. The medium was changed once on day 3. Astrocytes were used for experiments 7 days after the start of terminal differentiation. Stereological analysis was performed to determine the percentage of GFAP positive cells in each differentiation. Astrocyte cultures with higher than 85% GFAP positive cells were used for subsequent experiments.

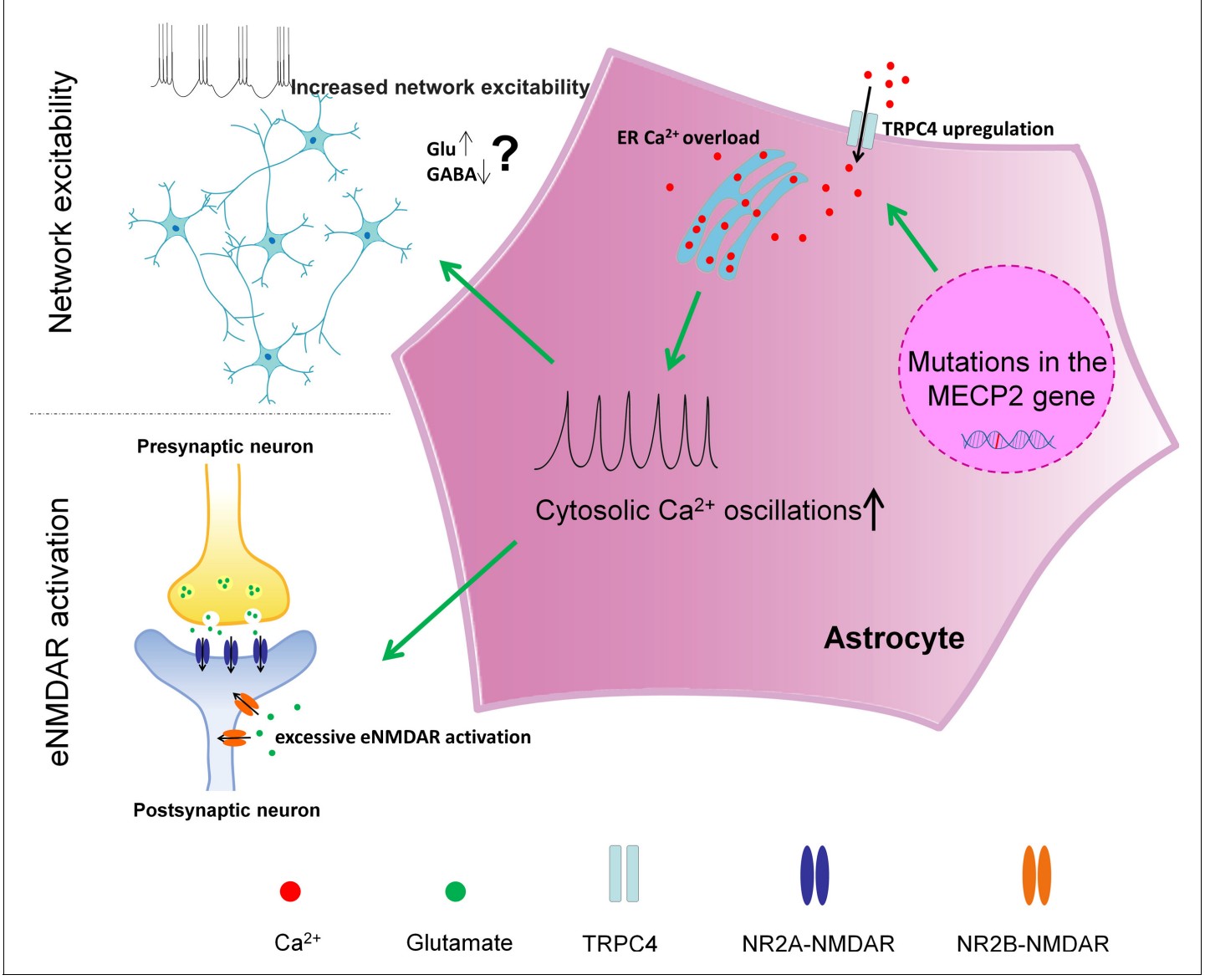

**Figure 9.** A schematic summary of the findings in our study.

DOI: https://doi.org/10.7554/eLife.33417.041

## Primary culture of mouse astrocytes

Mouse astrocyte cultures were prepared from postnatal day 0 (P0) male *Mecp2* null and wild type littermate mice, as well as P0 female *Mecp2$^{+/-}$* and *Mecp2$^{+/+}$* mice that carry an X chromosome-linked GFP transgene. After the brain of mice were removed and placed into ice-cold Hank's Balanced Salt Solution (HBSS), hippocampi were dissected and treated with 0.25% trypsin at 37°C for 30 min. Digestion was stopped by fetal bovine serum (FBS). Then the tissue was mechanically dissociated through a small fire-polished Pasteur pipette. After centrifuge, cells were re-suspended with DMEM, supplemented with penicillin/streptomycin and 10% FBS. Cells were plated into T-25 flasks and allowed to reach confluence. Contaminating neurons and microglia were then dislodged by shaking. The remaining adherent astrocytes were trypsinized, re-suspended in DMEM, and plated onto poly-D-lysine-coated coverslips at about 20,000 cells/ml. Cells were fed every 4 days by replacing the medium with fresh medium. The cells were used after 4 weeks in vitro. Stereological analysis was performed to determine the percentage of GFAP positive cells in each culture. Astrocyte cultures with higher than 90% GFAP positive cells were used for subsequent experiments.

## In vitro calcium imaging

For the human iPSC derived astrocytes and some mouse primary astrocytes, intracellular $Ca^{2+}$ was indicated by Fluo-4 or Rhod-2. Cells were bulk-loaded for 15 min at 37°C in artificial cerebrospinal fluid (aCSF) containing Fluo-4/AM (12.5 µg/ml), pluronic acid (0.05%), and DMSO (0.1%). After the $Ca^{2+}$ indicator was loaded, cells were transferred to a chamber and $Ca^{2+}$ imaging was performed. Mouse primary astrocytes from GCaMP6s mice were treated with 4-OH tamoxifen (0.5 µM, Sigma-Aldrich Inc.) for 24 hr to induce the expression of GCaMP6s. Cells were imaged using a Nikon A1 confocal microscope at room temperature. All image data were taken in the frame-scanning mode at 1 frame every 2 s (in the experiments using GCaMP6s as the $Ca^{2+}$ indicator, 1 frame every 1 s). Fluo-4 or GCaMP6s was excited at 488 nm. The $Ca^{2+}$ imaging data was analyzed using custom-written programs in Python. The metadata and the image data of the raw images were read with python-bioformats. $Ca^{2+}$ signals were presented as relative fluorescence changes ($\Delta F/F_0$) from specified regions of interest (ROIs). ROIs were selected using an automated segmentation algorithm (*Cai et al., 2016*), while somata and processes were manually identified on the basis of morphology. For the traces with baseline drift, baseline correction was performed using a 'rolling-ball' algorithm. The peaks were detected using the algorithm developed by Matlab (findpeaks function). The frequency, amplitude and the full-width at half maximum (FWHM) were calculated and measured. Images with obvious motion were excluded for analysis. In experiments that examined spontaneous $Ca^{2+}$ oscillations, the $Ca^{2+}$ level was reported as $F/F_0 = F_t/F_0$. Calcium elevation events were detected with thresholds of 3 times of standard deviation of the baseline. The source code for $Ca^{2+}$ events detection and automated segmentation is available in the supplementary files. Rhod-2 was used in the experiments involving viral expression of exogenous wild type MECP2, TRPC4, or shRNA against TRPC4. For the spontaneous astrocytic $Ca^{2+}$ activity imaging, aCSF containing (in mM): 120 NaCl, 3 KCl, 15 HEPES, 1 $MgCl_2$, 2 $CaCl_2$, 20 Glucose (pH7.4) was used. For TG-induced ER $Ca^{2+}$ release, aCSF with 0 mM $Ca^{2+}$ and 2 mM EGTA was used, and TG was perfused to induce the astrocytic $Ca^{2+}$ elevations. For SOCE, 2 mM $Ca^{2+}$ containing aCSF was perfused after TG induced ER $Ca^{2+}$ depletion. The spontaneous $Ca^{2+}$ elevations from the soma and process of astrocytes were analyzed separately. For the analysis of $Ca^{2+}$ imaging data in acute mouse brain slices, only $Ca^{2+}$ elevations in soma and major process of astrocytes were included.

## In vivo $Ca^{2+}$ imaging

GCaMP6s mice (Ai96, 024106) were purchased from Jackson Laboratory. GCaMP6s and GCaMP6s; $MeCP2^{flox/y}$ mouse pups were injected with AAV-GFAP-Cre-mCherry (see Materials and methods/ Adeno associated virus (AAV) injection into neonatal mice). Mice were used for $Ca^{2+}$ imaging after they are 2 months old. A cranial window (3 mm x 3 mm) was installed over the M2 frontal cortex on the left hemisphere of mutant and control mice (center coordinates: 1.5 mm anterior to Bregma, 1.0 mm lateral to midline). Mice were anesthetized with isoflurane during surgery and recovered to full awakeness (>45 min post surgery) before imaging layer I astrocyte calcium signals under the 25x water immersion lens (NA = 1.05) of a multi-photon microscope (FV1000MPE, Olympus) (*Cao et al., 2013*; *Managò et al., 2016*). The excitation wavelength of the two-photon laser was set at 900 nm, and the power of the laser emitted from the objective was set at 80 mW or lower. Image stacks had dimensions of 699 × 167 (width x length in µm) and the frame rate (frames per second) is 1.83. Three to five image recordings (655 s each) were made from each animal. The analysis of the in vivo $Ca^{2+}$ imaging data is described in Materials and methods/*In vitro* Calcium imaging. The traces were low-pass filtered (cut-off frequency: 0.5 Hz) before analysis to remove noises.

## Virus mediated expression of MECP2 and phenotypic analysis

Human or mouse astrocytes were infected with lentivirus encoding either GFP alone or GFP and wild type MECP2. 36 hr later, astrocytic $Ca^{2+}$ imaging was performed. Appropriate MOI was used to achieve infection rates higher than 90% in human astrocytes and ~50% in mouse astrocytes based on the percentage of GFP positive cells.

## Virus mediated expression of TRPC4

Human astrocytes were infected with lentivirus encoding either GFP alone or GFP and TRPC4. 10 days later, astrocytic $Ca^{2+}$ imaging was performed. The infection rate was higher than 80%, according to the percentage of GFP positive cells.

## Immunofluorescence

For immunohistological evaluation on brain sections, mice were killed by barbiturate overdose and perfused transcardially with phosphate-buffered saline (PBS), followed by 4% paraformaldehyde (PFA, in PBS). Brains were removed and post-fixed overnight. Then the brains were cryoprotected in buffered 30% sucrose (wt/vol) for at least 2 d. 25 µm coronal frozen sections were prepared using a cryostat microtome. For immunocytochemistry, cultures were fixed on ice with 4% PFA for 30 min. Immunostaining was performed as previously described (*Li et al., 2011*). Briefly, cultures were fixed on ice with 4% PFA for 30 min. After washed three times with PBS, cultures or sections were permeablized with 1% Triton X100 (Sigma, in PBS) for 30 min, blocked with 3% normal donkey serum and 0.25% Triton X100 in PBS for 90 min at room temperature (RT), and then incubated with primary antibodies overnight at 4°C. Then the cultures were incubated with the corresponding secondary antibodies for 60 min at RT, and washed five times with PBS at RT. Primary antibody dilutions were as follows: anti-GFAP (Millipore MAB3402 RRID:AB_94844, 1:500; and Dako, Z0334 RRID:AB_10013382, 1:500), anti-MeCP2 (Abcam ab50005 RRID:AB_881466, 1:500 and ab2828 RRID:AB_2143853, 1:500; and Cell Signaling Technology Cat# 3456S, RRID:AB_10828482, 1:500), anti-NeuN (Millipore MAB377 RRID:AB_2298772, 1:100), anti-S100β (Abcam Cat# ab868, RRID:AB_306716, 1:500) anti-TRPC4 (Alomone labs ACC-018 RRID:AB_2040239, 1:100; and Abcam ab84813 RRID: AB_1860087, 1:500). Secondary antibody dilutions were as follows: Alexa Fluor 568 Donkey-anti-Rabbit antibody (Thermo Fisher Scientific A10042 RRID:AB_2534017, 1:500), Alexa Fluor 488 Donkey-anti-mouse antibody (Thermo Fisher Scientific A21202 RRID:AB_141607, 1:500), and Alexa Fluor 647 Donkey-anti-mouse antibody (Thermo Fisher Scientific A-31571 RRID:AB_162542, 1:500). DAPI was used at 3 nM for counterstaining. Images were taken using a Nikon A1 confocal microscope.

## Western blot analysis

Astrocytes were collected into an Eppendorf tube and directly lysed by adding 1X LDS buffer (Life technology). Samples were sonicated to facilitate cell lysis and boiled for 5 min before loading into 10% SDS-PAGE gel. Proteins were transferred onto nitrocellulose membrane (Whatman) using a semi-dry transfer system from BioRad. Membrane was first blocked with 5% milk solution for 1 hr and then incubated with anti-TRPC4 (Alomone labs, ACC-018 RRID:AB_2040239, 1:100) diluted in 3% BSA solution at 4°C overnight. After incubating with infrared dye-conjugated secondary antibody (Thermo Fisher Scientific Cat# 35518 RRID:AB_614942; # SA5-35571 RRID:AB_2556775;1:10,000) for 1 hr at room temperature, membrane was scanned in an Odyssey infrared imaging system.

## RNA-Sequencing analysis

Total RNA was extracted from R294X-WT and R294X-MT iPSC-derived astrocytes using Trizol (life technology) and cleaned up with on-column DNase treatment (Qiagen). 150 ng total RNA was used to prepare sequencing library in a Mondrian SP +Workstation according to manufacturer's instructions (Nugen Encore SP +Complete). 100 bp single-end reads sequencing for each library was performed in an Illumina Hi-Seq 2000. Reads were mapped to the human genome build hg19 using Tophat (2.0.8) and differential gene expression analysis was done using Cuffdiff (2.0.2).

## Microarray analysis

Purified RNA was sent to Affymetrix for microarray analysis. HTA-2.0 chip was used in this project. Sample processing, dye labeling, hybridization, and scanning were performed by Affymetrix. Each sample was hybridized to four chips. Spike-in RNA was supplemented to samples and used as a normalization control.

## Chromatin immunoprecipitation (ChIP) and quantitative PCR (qPCR) to determine MeCP2 occupancy at the *Trpc4* gene promoter

Primary astrocyte culture was prepared from the forebrain of the *Mecp2-Flag* mice (*Li et al., 2011*) and control mice. ChIP was performed as previously described (*Li et al., 2011*). qPCR was done using a StepOne plus (Life technology). The following primers were used in the qPCR. TRPC4-P-1 forward: 5′-GCAGAGTGAGCCTGAGTCTA-3′. TRPC4-P-1 reverse: 5′-CGTGATCTCAAGACCAAGGG−3′. TRPC4-P-2 forward: 5′-TAGTATGGTTGGAGCAGGGC-3′. TRPC4-P-2 reverse: 5′-AGCTAAG TGGTGGTCAGGAC-3′. TRPC4-P-3 forward: 5′-CACCTTGGGAACGCAACTTT-3′. TRPC4-P-3 reverse: 5′-AAAACCCGCACGAAACCAG-3′. TRPC4-P-4 forward: 5′-CCCCATCGGAACTGACCA-3′. TRPC4-P-4 reverse: 5′-AGTATCCCAGATGTGAGGCC-3′.

## TRPC4 knockdown experiment

The hTRPC4 shRNA sequence used in our study is identical to that used in a previous study (*Zagranichnaya et al., 2005*). The shRNA sequence was cloned into a lentiviral construct, pLentilox3.7. Lentiviral particles were prepared as previously described (*Ananiev et al., 2011*). Mutant human astrocytes were infected with lentivirus encoding either GFP alone or GFP and shTRPC4. 36 hr later, astrocytic $Ca^{2+}$ imaging was performed. Appropriate MOI was used to achieve an infection rate of higher than 90% based on the percentage of GFP positive cells.

## Adeno-associated virus (AAV) injection into neonatal mice

Experiments described below were performed according to protocols approved by the IACUC at University of Wisconsin-Madison. Neonatal *Mecp2*$^{flox/y}$ mice were anesthetized by hypothermia before the injection. The AAV solution was injected into the lateral ventricles using a 5 µl Hamilton syringe. The injection site was located three-fifths in the line defined between the lambda intersection of the skull and each eye. The needle was held perpendicular to the skull surface and inserted to a depth of ∼3 mm. Then 1 µl of viral solution was injected into the ventricle. After injection, mice were transferred to a heated pad with perfused water at 38°C until they regained normal color and resumed movement. Three types of AAV were injected in this series of experiments: AAV8-hGFAP-mCherry-Cre (expression of fluorescence marker mCherry and the Cre recombinase under the astrocyte-specific hGFAP promoter), AAV8-Syn-mCherry-Cre (expression of mCherry and Cre under the neuron-specific human synapsin1 promoter), and AAV-mCherry. Stereological studies were performed to examine the infection rate of the AAV and the cell type specificity of mCherry expression and Cre-mediated deletion of MeCP2. Brain sections were immunostained with either anti-GFAP or anti-NeuN antibody and counterstained with DAPI; or combination of anti-GFAP and anti-MeCP2 antibodies; or combinations of anti-NeuN and anti-MeCP2 antibodies. Sections were viewed with a Zeiss photomicroscope at 20X. Stereology was performed with the StereoInvestigator software (MicroBrightField). The optimal number of counting sites was determined empirically based upon a Scheaffer CE value less than 0.30.

## Acute brain slices preparation and $Ca^{2+}$ imaging

Male mice at postnatal 2–3 weeks were anesthetized and coronal brain slices (400 µm) were prepared in ice-cold modified artificial cerebrospinal fluid (aCSF) (in mM: 124 NaCl, 2.5 KCl, 0.5 $CaCl_2$, 5 $MgCl_2$, 1.25 $NaH_2PO_4$, 26 $NaHCO_3$, and 15 glucose) bubbled with 95%$O_2$/5%$CO_2$. Then the slices were incubated in normal aCSF (in mM: 124 NaCl, 2.5 KCl, 2.5 $CaCl_2$, 1.2 $MgCl_2$, 1.25 $NaH_2PO_4$, 26 $NaHCO_3$, and 15 glucose) at room temperature for at least 1 hr. After incubation, the slices were bulk-loaded for 1 hr at 37°C in aCSF containing Fluo-4/AM (12.5 µg/ml), pluronic acid (0.05%), and DMSO (0.1%) saturated with 95%$O_2$/5%$CO_2$. Sulforhodamine 101 (SR101) was co-loaded as an astrocyte marker. Then slices were transferred to a chamber perfused with 95%$O_2$/5%$CO_2$ saturated aCSF. Slices were imaged using Nikon A1 confocal microscope. Fluo-4 and SR101 were excited at 488 nm and 561 nm respectively. The data acquisition protocol was the same as that used in cultured cells. The person performing the $Ca^{2+}$ imaging experiments and the subsequent data analysis was blind to the genotypes of mice used in these experiments.

## Electrophysiological recording

Slices were transferred to a recording chamber perfused with 95%$O_2$/5%$CO_2$ saturated aCSF. To record the extrasynaptic NMDA receptor current, hippocampal CA1 neurons were clamped at − 30

**Table 5.** List of statistical methods

| Figure | Statistical methods |
|---|---|
| *Figure 1C* | Mann-Whitney U Test |
| *Figure 1E* | Mann-Whitney U Test |
| *Figure 2C* | Two Tailed Unpaired t-Test |
| | Mann-Whitney U Test |
| | Mann-Whitney U Test |
| *Figure 2E* | Mann-Whitney U Test |
| *Figure 3C* | Two Tailed Unpaired t-Test |
| | Mann-Whitney U Test |
| *Figure 3E* | Mann-Whitney U Test |
| *Figure 4C* | Mann-Whitney U Test |
| *Figure 4E* | Mann-Whitney U Test |
| *Figure 5C* | Two Tailed Unpaired t-Test |
| | Mann-Whitney U Test |
| *Figure 6A* | Mann-Whitney U Test |
| *Figure 6C* | Mann-Whitney U Test |
| *Figure 6E* | Two Tailed Unpaired t-Test |
| *Figure 6F* | Two Tailed Unpaired t-Test |
| *Figure 6I* | Two Tailed Unpaired t-Test |
| *Figure 6K* | Two Tailed Unpaired t-Test |
| *Figure 6M* | Mann-Whitney U Test |
| *Figure 7C* | Mann-Whitney U Test |
| *Figure 7H* | Mann-Whitney U Test |
| *Figure 7L* | Two Tailed Unpaired t-Test |
| | Mann-Whitney U Test |
| | Mann-Whitney U Test |
| *Figure 8A* | Two Tailed Unpaired t-Test |
| *Figure 8B* | Mann-Whitney U Test |
| *Figure 8C* | Mann-Whitney U Test |
| *Figure 8E* | One-way ANOVA followed by post-hoc Holm-Sidak test |
| *Figure 8F* | One-way ANOVA followed by post-hoc Holm-Sidak test |
| *Figure 8H* | Two Tailed Unpaired t-Test |

DOI: https://doi.org/10.7554/eLife.33417.042

mV to relieve $Mg^{2+}$ block. We adjusted the equilibrium potential of $Cl^-$ to $-30$ mV by controlling the $[Cl^-]$ in the pipette solution. QX-314 was added to block $Na^+$ channels. The intracellular solution was (in mM): 95 Cs-methanesulfonate, 34 CsCl, 10 HEPES, 10 QX-314, 2.5 $MgCl2$, 1 $CaCl_2$, 5 NaCl, and 10 EGTA, pH 7.2. To selectively inhibit the astrocytic $Ca^{2+}$ activity, membrane-impermeable BAPTA (10 mM) was injected into the CA1 stratum radiatum astrocytes by patch clamp. To avoid a leakage of BAPTA from the pipette before the formation of whole cell patch clamp, the BAPTA solution was backfilled with the standard intracellular solution. The distinctive electrophysiological characteristics such as a linear I/V relationship was obtained to confirm the type of cells being patched.

To record store operated $Ca^{2+}$ influx in astrocytes, we transferred the coverslip into a chamber fixed in an Olympus BX51WI upright microscope. The recording chamber was perfused with aCSF containing (in mM): 120 NaCl, 3 CsCl, 15 HEPES, 1 MgCl2, 2 CaCl2, 20 Glucose (pH7.4). The pipette solution contained (in mM): 130 Cs-gluconate, 10 CsCl, 10 HEPES, 1 EGTA, 4 Mg-ATP and 0.5 Li-GTP. After whole cell patch clamp formed, the SOCE currents ($I_{SOCE}$) were induced by switching the perfusing solution to aCSF containing 1 μM TG. ML204 (10 μM) was used to isolate ML204 sensitive currents.

The membrane capacitance was also obtained to normalize the currents. To get the I-V curve of $I_{SOCE}$, a voltage ramp from 100 mV to −100 mV (for 200 ms) was applied. The raw data was acquired with a Multiclamp 700B amplifier and pClamp10.2 software (Axon Instruments, Sunnyvale, CA). All data was analyzed using Clampfit10.2. The current density at −70 mV was quantified.

For the in vitro epilepsy model, male mice at postnatal 5–8 weeks were used. The slices were transferred to a recording chamber perfused with 95%$O_2$/5%$CO_2$ saturated aCSF containing 5 mM KCl. Epileptiform activity was induced by switching perfused solution to aCSF containing the $GABA_A$ receptor antagonist bicuculline (Bic, 3 µM; Sigma) (*McLeod et al. (2013)*. Reduced seizure threshold and altered network oscillatory properties in a mouse model of Rett syndrome. 231, 195–205.). Epileptiform events were detected with Clampfit 10.2 (Axon Instruments) and verified visually. The intracellular solution was (in mM): 140 K-gluconate, 7.5 KCl, 10 Hepes-K, 0.5 EGTA-K, 4 Mg-ATP and Li-GTP. The currents caused by unclamped action potentials has been truncated (*Fellin et al., 2006*).

## Statistics analysis

All data were analyzed using the SigmaPlot 13.0 (Systat Software, Inc RRID:SCR_003210). Student's t-tests or Mann–Whitney U tests were used to make comparisons between two groups unless indicated otherwise. One-way ANOVA followed by post-hoc Holm-Sidak test was performed when there are three or more groups. $Ca^{2+}$ imaging data was analyzed by two-way ANOVA. A list of the statistical methods used in each figure can be found in *Table 5*. Average data are shown as the mean ±SEM. p values < 0.05 were considered statistically significant. No statistical methods were used to pre-determine sample sizes. Data were collected from randomly selected fields within images. More than three independent experiments/biological replicates (for human astrocyte data set, each independent experiment was defined as one independent differentiation; for mouse astrocyte in vitro data set, each independent experiment was defined as one isolation culture from one mouse; for mouse in situ and in vivo data set, both number of cells and number of mice were described) were included in each data set.

## Acknowledgements

We thank Xiaoji Zhang for assistance with mouse work, and Dr. Dan Bolt for assistance with statistical analysis. RL was supported by a pre-doctoral fellowship from the Stem Cell and Regenerative Medicine Center at the University of Wisconsin-Madison. This work was partially supported by R21 NS081484, R56NS100024, and R01HD064743 to QC, ZIA MH002897 to KHW and QL, and U54HD090256 to the Waisman Center.

## Additional information

### Funding

| Funder | Grant reference number | Author |
| --- | --- | --- |
| National Institute of Neurological Disorders and Stroke | R21NS081484 | Qiang Chang |
| National Institute of Mental Health | ZIAMH002897 | Kuan Hong Wang |
| Eunice Kennedy Shriver National Institute of Child Health and Human Development | U54HD090256 | Qiang Chang |
| Eunice Kennedy Shriver National Institute of Child Health and Human Development | R01HD064743 | Qiang Chang |
| National Institute of Neurological Disorders and Stroke | R56NS100024 | Qiang Chang |

The funders had no role in study design, data collection and interpretation, or the decision to submit the work for publication.

## Author contributions
Qiping Dong, Data curation, Formal analysis, Investigation, Methodology, Writing—review and editing; Qing Liu, Data curation, Formal analysis, Investigation, Methodology; Ronghui Li, Anxin Wang, Qian Bu, Data curation, Investigation; Kuan Hong Wang, Supervision, Writing—review and editing; Qiang Chang, Conceptualization, Data curation, Supervision, Funding acquisition, Writing—original draft, Project administration, Writing—review and editing

## Author ORCIDs
Ronghui Li http://orcid.org/0000-0001-6329-5895
Kuan Hong Wang http://orcid.org/0000-0002-2249-5417
Qiang Chang http://orcid.org/0000-0002-7625-2170

## Ethics
Animal experimentation: This study was performed in strict accordance with the recommendations in the Guide for the Care and Use of Laboratory Animals of the National Institutes of Health. All of the animals were handled according to approved institutional animal care and use committee (IACUC) protocols (G005315) of the University of Wisconsin-Madison

## Decision letter and Author response
Decision letter https://doi.org/10.7554/eLife.33417.048
Author response https://doi.org/10.7554/eLife.33417.049

# Additional files
### Supplementary files
• Source code 1. Source code ca_signal.py: code used for the analysis of fluorescence changes, the detection of calcium events, measurement of peak amplitudes, and full width at half maximum in *Figures 1–8*.
DOI: https://doi.org/10.7554/eLife.33417.043

• Source code 2. Source code segment.py: code used for the automated segmentation step in image analysis in *Figures 1–8*. It identifies each ROI by the results of cross correlation analysis. Briefly, all the local maximum pixels are detected as the initial ROIs. Then the cross correlation of fluorescence change between an ROI and a surrounding pixel is calculated. If the cross correlation is greater than a given threshold, this pixel is added to the ROI. The procedure is repeated until no more pixel can be added.
DOI: https://doi.org/10.7554/eLife.33417.044

• Supplementary file 1. Expression levels of SOCE-related genes in wild type and mutant RTT human astrocytes.
DOI: https://doi.org/10.7554/eLife.33417.045

• Transparent reporting form
DOI: https://doi.org/10.7554/eLife.33417.046

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
