## [Decision Letter]

[Editors’ note: a previous version of this study was rejected after peer review, but the authors submitted for reconsideration. The first decision letter after peer review is shown below.]

Thank you for choosing to send your work, "Mechanism and Consequence of Abnormal Calcium Homeostasis in Rett Syndrome Astrocytes", for consideration at *eLife*. Your initial submission has been assessed by a Senior Editor in consultation with a member of the Board of Reviewing Editors. Although the work is of interest, we regret to inform you that the findings at this stage are too preliminary for further consideration at *eLife*.

Elucidating how astrocytes contribute to the pathobiology of Rett Syndrome is interesting and important question. The authors provide evidence that cell autonomous changes in astrocyte calcium signaling affect neurons in Rett model mice. However, there were several shared issues raised by reviewers that diminish enthusiasm for this paper. Specifically, more direct evidence that calcium signaling is altered in mecp2- null astrocytes in vivo is needed to strengthen their conclusions. In addition, the in vitro calcium imaging data require further controls and analyses to strengthen their conclusions. The use of Rett patient-derived IPSCs to study astrocyte dysfunction and the demonstration that similar effects are observed in mouse null astrocytes are strength; however more characterization of astrocytes and clarification of the protocols used to generate human astrocytes are needed. Reviewers raised additional concerns regarding the quality and consistency of the data that need to be addressed.

Reviewer #1:

The paper makes a compelling argument that cell autonomous changes in astrocyte calcium signaling affects neurons in RETT model mice. Key aspects of the calcium signaling studies are supported by work in human astrocyte cultures.

Overall, this is a valuable and useful study and adds to the growing body of work that shows astrocyte calcium signals are altered in neurological disease. However, several points should be addressed.

1) In most of the figures, the photomicrographs of images are rather poor and hard to make out. Replace them with better examples. If needed, adjust the gray levels so the reader can see the images and determine if what is stated in the text is supported by the data shown.

2) There are no data for evoked calcium signals in the paper. The authors have used GPCR agonist-mediated calcium signals. They should clarify this in the paper and explicitly state that evoked signals due to action potential firing were not studied.

3) I think the data for and R294X should be shown separately in a Table. I don't understand the logic of averaging data from two separate genotypes. Alternatively, they could show the data from just one genotype. Averaging across two distinct disease associated genotypes seems odd.

4) Why is it that astrocytes in culture (human and mouse) have only a small population of cells that show spontaneous calcium elevations? What is this proportion in the mouse slice studies? Why do only a small population of cells show calcium responses in WT and MT? Usually, all astrocytes in vivo show calcium signals and so the low proportion is odd. Please explain or discuss. Does this reflect an artifact of cell culture?

5) The RNAseq data are mentioned but not shown. These data seemingly exist and should be reported in the main text and figures. How else can the reader assess the quality of the data set and believe the findings with TRPC4? The RNAseq data deserve to be reported in full.

6) What is the molecular basis of the ML204-insensitive current in astrocytes? Please discuss or explain with experiments.

7). The data with hGFAP-Cre AAVs are not convincing as a way to target astrocytes selectively, especially in the hippocampus. My suggestion is to drop these data and use the space that is gained to report the RNAseq data in full. I don' think the hGFAP-Cre data add anything useful to the paper, because the logic for why this should be astrocyte specific is not clear.

Overall, a potentially valuable paper that needs another round of hard work to tighten up some aspects.

Reviewer #2:

Summary:

The authors of this manuscript attempt to prove that MeCP2 mutant astrocytes from both iPSC and mouse models display altered calcium homeostasis by a mechanism of high internal calcium store concentration and increased expression of TRPC4 channels. This leads to increased spontaneous calcium activity, increased amplitude to pharmacologically induced calcium transients, and a proposed aberrant activation of NMDA receptors. While this manuscript presents an interesting first attempt at probing calcium homeostasis in MeCP2 mutant astrocytes, there are significant and fundamental errors in both experiments and the approaches used that undermine the interpretation of results.

Essential revisions:

1) Throughout the paper, the authors claim to show in vivo evidence that supports their conclusions. There is not a single in vivo experiment in this entire data set. They consistently refer to their acute slice experiments as in vivo work: this is incorrect. Acute slice work is generally referred to as in situ, as it does not wholly replicate the in vivo environment. This is critically important when it comes to astrocytes, which are extremely sensitive to osmotic and ion changes that can occur during the process of slicing and can alter astrocyte activity.

2) There is no characterization of the iPSC cultures to determine if they are in fact astrocytes. A reference is given to a previous paper, but no characterization of the cultures used in the experiments conducted in this manuscript is provided. It is critical to include experiments where the cultures used for calcium imaging are fixed and stained for astrocyte markers.

3) Calcium imaging data: Overall, the calcium imaging data from both iPSCs, primary cultures and in situ slice experiments is not convincing. Fluo-4 is a single wavelength dye, and while normalizing to baseline can reduce issues concerning concentrations of indicator, this indicator can only be used for relative measures of calcium amplitudes and requires careful loading controls that do not appear to be presented. Use of a ratiometric dye for experiments would have been more convincing for the changes that are described. Additionally, in the entire manuscript the data presented is from somatic recordings of astrocyte transients. While for the iPSCs work this may be understandable, for the primary culture and slice work there should have been an effort to examine calcium transients in the enlarged cell body or processes of astrocytes, as the somatic activity of an astrocyte is incredibly low compared to the activity in processes (see Di Castro et al., 2011) and represent a small fraction of the total calcium activity. As the authors used a genetically encoded calcium indicator in some experiments, why not in all experiments? This would have been more convincing.

4) The frame rate for calcium imaging is too slow for an accurate recording of the spontaneous activity and is likely underestimating the frequency of spontaneous events.

5) In general, in the majority of the experiments, the n's presented for results are low for what this reviewer would expect from culture experiments where the dish contains many thousands of cells.

6) The choice of representative images, such as the immunostaining in Figure 5A, is inconsistent with the data presented. The authors use a lentivirus to reintroduce Mecp2 into their null cell populations, along with GFP. The MeCp2-null cells depicted by the GFP images seem to be more numerous and have different morphology. Furthermore, there seem to be some red signal indicating some expression of MeCP2 in these cells. This calls into question the potential health and fidelity of the cells, or the choice of images. A similar issue is found in Figure 5H: a representative staining has several cells from MeCP2-null neonates selected that express GFP and Rhod-2. However, a close examination of these areas does not show any Rhod-2, which could explain why the authors do not report any Ca transients in these regions. Again, the choice of representative images is called into question, since the representative traces depict absolutely no traces during a ~10 min period, yet their data show the frequency in this population to be 1.33 / 0.1 min-1. In general, throughout the paper the representative data seems inconsistent with the reported quantification.

7) The authors of the paper make an assumption concerning astrocyte calcium signaling that there have been no changes to the signaling pathways contributing to the generation of spontaneous signaling aside from TRPC4. This is highly unlikely given the nature of MeCP2. The major release pathway from the ER involves activation of IP3 receptors. While TRPC4 does appear altered, it may also be the case that other components of the ER release pathway may be altered as well and be the root cause of the increased spontaneous signals. A closer examination of other components of ER associated calcium signaling should be conducted. Additionally, staining for TRPC4 to show it is even expressed by astrocytes in vivo is needed to validate the model.

Reviewer #3:

The manuscript by Qiping Dong et al., provides in vitro and ex vivo data that astrocytes from Rett Syndrome patients and mouse models of the human disorder have cell-autonomous increases in cytosolic calcium oscillations. Furthermore, data suggest that increased astrocytic calcium responses results in increased activation of extrasynaptic NMDA receptors on neighboring neurons. The authors identify that increased cytosolic calcium in astrocytes is primarily derived from the endoplasmic reticulum through increased store operated calcium entry (SOCE). They also identify a potential mechanism for increased SOCE in mutant and null astroctyes via increased levels ofTRPC4. The authors postulate that this increased astrocytic calcium leads to seizures in Rett patients via increased activation of extrasynaptic NMDA receptors.

Identifying how astrocytes play a role in Rett Syndrome is interesting, timely, and therapeutically relevant. The use of Rett patient-derived IPSCs to study astrocyte dysfunction is innovative and impactful and the demonstration that similar effects are observed in mouse null astrocytes is rigorous. However, there are some major concerns regarding the quality and consistency of the data as well as the relevance for effects in vivo. See points below.

1) All data are in vitro (cultured cells) or ex vivo (slice) with no in vivo evidence that astrocytic calcium dynamics and subsequent extrasynaptic NMDA transmission are dysregulated in Rett Syndrome. While it is appreciated that not everyone is capable of imaging calcium oscillations in vivo, one relatively straight forward experiment that would start to address this point is to assess seizure susceptibility in vivo (vs. ex vivo) following astrocyte-specific knockdown or ablation of Mecp2.

2) There are multiple inconsistencies in the data reporting changes in calcium in mutant or null astrocytes versus wild-type. Often calcium dynamics are very different across figures for the same genetic or pharmacological manipulation. For example, the calcium dynamics reported in Figure 12C,D show different levels of mutant calcium oscillation and frequency than is reported in Figure 2B,C. These discrepancies need to be addressed.

3) The authors fail to clearly define their N numbers in their figures or results. It is not clear from reading the manuscript whether the N's provided refer to number of cells, number of experiments, etc. Given that the N's are relatively high, one would assume this represents cells. However, it is unclear how many independent experiments these N's represent. Data should be assessed from at least three independent experiments and the authors should sample similar numbers of cells across experiments. This information should be clearly outlined. Furthermore, while the authors list N's in the text, the authors should list this in each figure legend for clarity. The authors also need to list the stats and p-values for each experiment in the figure legend. This is inconsistent across the manuscript.

4) The authors conclude that evoked calcium activities are disrupted following either ATP or glutamate stimulation. The mutant or null astrocytes already have higher amplitude responses spontaneously (e.g. Figure 1E). If you compare the amplitudes of spontaneous responses to evoked responses, it appears that the actual magnitude of increase following stimulation is not different between wild-type and null or mutants astrocytes (e.g. Figure 1-amplitude increases from about 2 (spontaneous) to about 3.5 (+ATP) for WT and from about 3 (spontaneous) to about 4.5 (evoked) in MT). Therefore, the conclusion that evoked responses are different is not accurate and should be clarified.

5) The figures are generally very difficult to follow and could use some reorganization. For example, the authors often go back and forth between human and mouse within the same figure without clearly identifying this within the figure. In addition, a number of figures could be either combined or included as supplemental material. For example, Figure 8, Figure 9, Figure 10 and Figure 11 could all be represented along with data from Figure 7 or included as a supplemental file.

6) The images presented throughout the manuscript often lack many general cell markers that would benefit interpretation of the data. In some cases, the figure lacks any images of cultures (Figure 2). Most images do not include DAPI to label all cells and no images show staining for astrocyte-specific markers. Higher magnification images should also be added to the current low magnification images. In addition, the authors list quantification in the text but these data should be included in the figure to increase clarity.

7) The authors need to validate sufficient knockdown of genes and reduced protein in astrocytes following shRNAs. These data can be included as supplemental.

8) In Figure 13 and Figure 14, there is only quantification of mCherry labeling not loss of Mecp2 expression. The authors must quantify presence or absence of Mecp2 protein in mCherry positive and negative astrocytes and neurons. This is particularly important given that detecting Mecp2 in vivo in non-neuronal cells is notoriously difficult and typically requires amplification (Ballas et al., 2011). From the images, it is not clear that the antibody recognizes Mecp2 in uninfected astrocytes, which is important to show in order to validate loss of Mecp2 in infected astrocytes. This figure could also benefit from higher magnification images. Lastly, the authors suggest that they are creating cell-specific knockout mice. This is exaggerated as only about 27% of cells are infected.

9) The manuscript emphasizes that their heightened extrasynaptic NMDA receptor activation has never before been reported in any RTT models. This is inaccurate. There is at least one recent paper reporting increased activation of eNMDARs in RTT (Lo et al., 2016). While it is appreciated that this effect has never been attributed to astrocytes, the text should be modified to include these published findings.

10) The model provided in Figure 15 is not detailed enough to add any meaningful contribution to the manuscript as a whole and could be excluded.

[Editors’ note: what now follows is the decision letter after the authors submitted for further consideration.]

Thank you for submitting your article "Mechanism and Consequence of Abnormal Calcium Homeostasis in Rett Syndrome Astrocytes" for consideration by *eLife*. Your article has been reviewed by three peer reviewers, and the evaluation has been overseen by a Reviewing Editor and a Senior Editor. The reviewers have opted to remain anonymous.

The reviewers have discussed the reviews with one another and the Reviewing Editor has drafted this decision to help you prepare a revised submission.

Summary:

The authors have made significant improvements to their manuscript from the original submission. The authors have addressed several of the key concerns raised in the first submission. The revised manuscript is better organized, clear in its presentation of the data and the model systems used and draws well-grounded conclusions from the data. However, there are a number of additional revisions and comments that will need to be addressed before publications as summarized below.

Essential revisions:

1) Most data supporting the TRPC4 mechanism are in vitro. The only data showing some evidence of increased TRPC4 function in Mecp2 null mice are immunostaining in Figure S10. More data demonstrating TRPC4 expression changes and function in vivo/ex vivo would strengthen the study. For example, western blots for TRPC4 from acutely isolated astrocytes (mouse model) in addition cultured astrocytes would be helpful.

At minimum, please include in brief discussion emphasizing the need for future studies on the topic. This would help the reader and thoughtfully point out the limitations of the current work.

2) The authors' explanation to reviewers regarding the large variability in N's is not sufficient. In many cases the N's vary by 100s of cells. For example, Figure 1C the N=526 cells for mutant cultures and N=811 for wild-type cultures. As one reviewer previously pointed out, this could greatly affect the statistical analyses. Further, it is unclear how the authors chose a particular N prior to doing statistics and it remains unclear in the manuscript how many independent experiments each data set represents.

3) While the authors make more attempt to validate their cell-specific genetic manipulations of Mecp2, these data are still not quantified sufficiently. The data in Supplementary file 3 and Figure 8—figure supplement 2 only shows quantification of% cells with mCherry, GFAP, and NeuN expression. The% of NeuN+ and GFAP+ cells +/- MECP2 immunoreactivity after AAV injection should also be included here. Given that the goal is to achieve cell-type specific manipulation, these data are important.

4) Many immunofluorescence images are still generally low quality. For example, in Figure 6—figure supplement 2, the background staining of TRPC4 is quite a bit higher in MECP2 null animals vs. WT. The non-specific background staining should be the same in these images to ensure a more accurate comparison. This and other figures would also benefit from some higher magnification zoom-ins in addition to the low magnification images. There is also immunostaining that is very difficult to visualize in several figures. One clear example is Figure 8—figure supplement 2 (MECP2 and NeuN immunoreactivity).

---

## [Author Response]

[Editors’ note: the author responses to the first round of peer review follow.]

Reviewer #1:The paper makes a compelling argument that cell autonomous changes in astrocyte calcium signaling affects neurons in RETT model mice. Key aspects of the calcium signaling studies are supported by work in human astrocyte cultures.Overall, this is a valuable and useful study and adds to the growing body of work that shows astrocyte calcium signals are altered in neurological disease. However, several points should be addressed.

We are grateful the reviewer thinks our study is valuable and makes a compelling argument.

1) In most of the figures, the photomicrographs of images are rather poor and hard to make out. Replace them with better examples. If needed, adjust the gray levels so the reader can see the images and determine if what is stated in the text is supported by the data shown.

We have replaced all images with better ones.

2) There are no data for evoked calcium signals in the paper. The authors have used GPCR agonist-mediated calcium signals. They should clarify this in the paper and explicitly state that evoked signals due to action potential firing were not studied.

We have used the terms “pharmacologically evoked” and “agonist evoked” to specify ATP- and glutamate-evoked calcium elevation and distinguish them from those evoked by neuronal activity under physiological conditions. Furthermore, those assays involving evoked activities were only meant to be additional measure of altered calcium homeostasis, but not to imply any underlying mechanism.

3) I think the data for V247f and R294X should be shown separately in a Table. I don't understand the logic of averaging data from two separate genotypes. Alternatively, they could show the data from just one genotype. Averaging across two distinct disease associated genotypes seems odd.

We combined data from the two genotypes, because both appears to be null at the protein level. Nonetheless, we have taken the reviewer’s suggestion by presenting the R294X (a common RTT mutation) genotype in the main text and mentioned the R247fs genotype in supplementary data in our revised manuscript.

4) Why is it that astrocytes in culture (human and mouse) have only a small population of cells that show spontaneous calcium elevations? What is this proportion in the mouse slice studies? Why do only a small population of cells show calcium responses in WT and MT? Usually, all astrocytes in vivo show calcium signals and so the low proportion is odd. Please explain or discuss. Does this reflect an artifact of cell culture?

Our calcium imaging results represent a snapshot sampling of the cell population during a relatively short time window (10 minutes). The higher percentage of cells showing spontaneous calcium activity in MT means a higher chance than in WT. If the imaging period is longer, the percentage will increase in both genotypes, yet the percentage will still be higher in the MT.

5) The RNAseq data are mentioned but not shown. These data seemingly exist and should be reported in the main text and figures. How else can the reader assess the quality of the data set and believe the findings with TRPC4? The RNAseq data deserve to be reported in full.

We are committed to deposit our RNA-seq data in public database after our manuscript is accepted for publication. Since the whole RNA-seq data set is not the main focus of our study, we have decided to describe its role as the discovery approach leading our attention to TRPC4. In order to bolster the validity of our conclusion of increased TRPC4 expression, we added data from multiple independent experimental approaches, including microarray, qRT-PCR, Western blot, and immunostaining on brain sections. Results from all of those experiments consistently showed elevated TRPC4 expression in MeCP2 mutant astrocytes.

6) What is the molecular basis of the ML204-insensitive current in astrocytes? Please discuss or explain with experiments.

Depleting ER calcium leads to the activation of store operated calcium entry. In astrocytes, this calcium entry may be mediated by TRP channels and calcium release activated calcium channels (Oral1). ML204 is a newly discovered inhibitor that only blocks TRPC4 containing channels but not Oral1 channels mediated currents. Thus, we used the ML204-sensitive current as an indirect measurement of TRPC4 mediated current. An increase of ML204-sensitive current (i.e. TRPC4 current) is consistent with our findings of increased TRPC4 expression in MT astrocytes.

7) The data with hGFAP-Cre AAVs are not convincing as a way to target astrocytes selectively, especially in the hippocampus. My suggestion is to drop these data and use the space that is gained to report the RNAseq data in full. I don' think the hGFAP-Cre data add anything useful to the paper, because the logic for why this should be astrocyte specific is not clear.

In our study, there are two sets of experiments involving hGFAP-Cre AAV injection. In the first set, our analysis of phenotypes (both eNMDAR activation and calcium dynamics) was performed in acute hippocampal slices prepared from 2-3-week-old mice (Figure 8). Even if neural stem cells in the hippocampus infected with AAV express hGFAP-cre, there won't be enough time (2 weeks after injection of the AAV into postnatal day 0 pups) for those neural stem cells to differentiate into neurons and complicate our experimental results. Indeed, when we further sectioned and stained those slice, 100% of the mCherry positive cells in those sections were GFAP positive astrocytes (Figure 8 and Table 3).

The second set is our imaging of spontaneous calcium activity in live mice. Although the lapse of time after AAV administration was longer, our analysis was limited to the superficial layer (layer 1) of the frontal cortex, in which region our immunological analysis again confirmed the specificity of the AAV-cre mediated MeCP2 deletion was limited to astrocytes.

Overall, a potentially valuable paper that needs another round of hard work to tighten up some aspects.Reviewer #2:Essential revisions:1) Throughout the paper, the authors claim to show in vivo evidence that supports their conclusions. There is not a single in vivo experiment in this entire data set. They consistently refer to their acute slice experiments as in vivo work: this is incorrect. Acute slice work is generally referred to as in situ, as it does not wholly replicate the in vivo environment. This is critically important when it comes to astrocytes, which are extremely sensitive to osmotic and ion changes that can occur during the process of slicing and can alter astrocyte activity.

We have revised our description of the slice results by using the phrase “in situ” instead. In addition, we have collaborated with the Wang laboratory at NIMH, an expert in live imaging, to perform calcium imaging experiment in live mice, which replicate the in vivo environment. Consistent with our findings in cultured human and mouse astrocytes and mouse brain slices, we observed increased frequency of spontaneous calcium activity in astrocytes in live mice whose *Mecp2* gene was predominantly deleted in cortical astrocytes. These new data are included in our revised manuscript (Figure 5).

2) There is no characterization of the iPSC cultures to determine if they are in fact astrocytes. A reference is given to a previous paper, but no characterization of the cultures used in the experiments conducted in this manuscript is provided. It is critical to include experiments where the cultures used for calcium imaging are fixed and stained for astrocyte markers.

We have performed detailed characterization of astrocytes differentiated from congenic pairs of Rett syndrome iPSC lines in our previous publication (Williams et al., 2014), and again in our current study. We didn’t include those data in our original submission but have included them in our revised manuscript (Figure 1—figure supplement 1).

3) Calcium imaging data: Overall, the calcium imaging data from both iPSCs, primary cultures and in situ slice experiments is not convincing. Fluo-4 is a single wavelength dye, and while normalizing to baseline can reduce issues concerning concentrations of indicator, this indicator can only be used for relative measures of calcium amplitudes and requires careful loading controls that do not appear to be presented. Use of a ratiometric dye for experiments would have been more convincing for the changes that are described. Additionally, in the entire manuscript the data presented is from somatic recordings of astrocyte transients. While for the iPSCs work this may be understandable, for the primary culture and slice work there should have been an effort to examine calcium transients in the enlarged cell body or processes of astrocytes, as the somatic activity of an astrocyte is incredibly low compared to the activity in processes (see Di Castro et al., 2011) and represent a small fraction of the total calcium activity. As the authors used a genetically encoded calcium indicator in some experiments, why not in all experiments? This would have been more convincing.

In our experiments, we focus more on the relative changes of intracellular calcium concentration rather than the absolute concentration. In addition, our focus was on the comparison between the MT and WT genotypes. Finally, the frequency results should not be affected by the loading conditions.

That being said, we took the reviewer’s suggestion, and performed additional experiments while including calcium ionophore A23187 as a loading control (Figure 1—figure supplement 2) in calcium imaging experiments involving human astrocytes. More importantly, we used the GCaMP6 reporter mice in calcium imaging experiments in primary mouse astrocytes (Figure 3) and live mice (Figure 5), both of which yielded results consistent with calcium dye-based results from cultured human astrocytes (Figure 1) and acute mouse brain slices (Figure 4). Finally, we took the reviewer’s suggestion to re-analyze the calcium activity of astrocytic processes and presented them separately (Figure 1—figure supplement 3).

4) The frame rate for calcium imaging is too slow for an accurate recording of the spontaneous activity and is likely underestimating the frequency of spontaneous events.

To avoid potential phototoxicity, we did calcium imaging at the rate of one image per every 2 seconds. The kinetics of astrocytic calcium activity is far slower than this frame rate. We did the kinetics analysis and the FWHM (full width at half maximum) is ~40 Sec for soma and ~30 sec for processes. Also, there are many papers in which the authors used similar sampling rate. For GCaMP6s imaging, we used faster frame rates (mouse primary astrocytes (Figure 3): 1 frame per second; live mice (Figure 5): 1.83 frame per second).

5) In general, in the majority of the experiments, the n's presented for results are low for what this reviewer would expect from culture experiments where the dish contains many thousands of cells.

During the short imaging period (10 minutes) for each imaging field, only a percentage of the astrocytes in culture exhibited spontaneous calcium elevations. Only these cells were included in the statistical analysis. Further, being mindful that long duration of such imaging experiment may affect the health of cells and the data consistency, we limited the total amount of imaging time for each dish, thereby obtaining only a few images from a single dish. We hope the consistency in results between species and across the in vitro, in situ, and in vivo platforms will alleviate the reviewer’s concern.

6) The choice of representative images, such as the immunostaining in Figure 5A, is inconsistent with the data presented. The authors use a lentivirus to reintroduce Mecp2 into their null cell populations, along with GFP. The MeCp2-null cells depicted by the GFP images seem to be more numerous and have different morphology. Furthermore, there seem to be some red signal indicating some expression of MeCP2 in these cells. This calls into question the potential health and fidelity of the cells, or the choice of images. A similar issue is found in Figure 5H: a representative staining has several cells from MeCP2-null neonates selected that express GFP and Rhod-2. However, a close examination of these areas does not show any Rhod-2, which could explain why the authors do not report any Ca transients in these regions. Again, the choice of representative images is called into question, since the representative traces depict absolutely no traces during a ~10 min period, yet their data show the frequency in this population to be 1.33 / 0.1 min-1. In general, throughout the paper the representative data seems inconsistent with the reported quantification.

We apologize for the poor quality of the representative images, which was due to image processing during the submission of the manuscript. We have corrected that issue. In new images included in the revised manuscript, Figure 2A shows no difference in cell morphology between mutant and wild type. Finally, in the mouse astrocytes rescue experiments presented in Figure 5 of our original submission, the average of frequency was calculated using cells showing calcium transient. In another word, although all cells (cell 1-8) were used to calculate percentage of cells with calcium activity, only cells with calcium activity (cell 1, 5, 6, 7, 8) were used to calculate frequency and amplitude. Given the data are largely redundant with those from the human cells, we removed them from our revised manuscript as part of our effort to streamline the organization.

7) The authors of the paper make an assumption concerning astrocyte calcium signaling that there have been no changes to the signaling pathways contributing to the generation of spontaneous signaling aside from TRPC4. This is highly unlikely given the nature of MeCP2. The major release pathway from the ER involves activation of IP3 receptors. While TRPC4 does appear altered, it may also be the case that other components of the ER release pathway may be altered as well and be the root cause of the increased spontaneous signals. A closer examination of other components of ER associated calcium signaling should be conducted. Additionally, staining for TRPC4 to show it is even expressed by astrocytes in vivo is needed to validate the model.

We agree that additional components of the calcium homeostasis regulation may be affected, and plan to further investigate in the future. As suggested by the reviewer, we performed immunostaining of Trpc4 on brain sections from wild type and *Mecp2* knockout mice and observed increased intensity of Trpc4 immunoreactivity in astrocytes in brain sections from the *Mecp2* knockout mice. These new results are consistent with our observation of increased TRPC4/Trpc4 expression in cultured human and mouse astrocytes, and are included in the revised manuscript (Figure 6—figure supplement 2).

Reviewer #3:1) All data are in vitro (cultured cells) or ex vivo (slice) with no in vivo evidence that astrocytic calcium dynamics and subsequent extrasynaptic NMDA transmission are dysregulated in Rett Syndrome. While it is appreciated that not everyone is capable of imaging calcium oscillations in vivo, one relatively straight forward experiment that would start to address this point is to assess seizure susceptibility in vivo (vs. ex vivo) following astrocyte-specific knockdown or ablation of Mecp2.

We have collaborated with the Wang laboratory at NIMH to perform calcium imaging experiment in live mice, which replicate the in vivo environment. Consistent with our findings in cultured human and mouse astrocytes and mouse brain slices, we observed increased frequency and amplitude of spontaneous calcium activity in astrocytes in astrocyte-specific *Mecp2* knockout mice. These new data are included in our revised manuscript (Figure 5).

2) There are multiple inconsistencies in the data reporting changes in calcium in mutant or null astrocytes versus wild-type. Often calcium dynamics are very different across figures for the same genetic or pharmacological manipulation. For example, the calcium dynamics reported in Figure 12C,D show different levels of mutant calcium oscillation and frequency than is reported in Figure 2B,C. These discrepancies need to be addressed.

The apparent inconsistency is mostly due to poor selection of sample traces. We have included more representative sample traces in our revised manuscript.

3) The authors fail to clearly define their N numbers in their figures or results. It is not clear from reading the manuscript whether the N's provided refer to number of cells, number of experiments, etc. Given that the N's are relatively high, one would assume this represents cells. However, it is unclear how many independent experiments these N's represent. Data should be assessed from at least three independent experiments and the authors should sample similar numbers of cells across experiments. This information should be clearly outlined. Furthermore, while the authors list N's in the text, the authors should list this in each figure legend for clarity. The authors also need to list the stats and p-values for each experiment in the figure legend. This is inconsistent across the manuscript.

We will include the requested information in our revised manuscript.

4) The authors conclude that evoked calcium activities are disrupted following either ATP or glutamate stimulation. The mutant or null astrocytes already have higher amplitude responses spontaneously (e.g. Figure 1E). If you compare the amplitudes of spontaneous responses to evoked responses, it appears that the actual magnitude of increase following stimulation is not different between wild-type and null or mutants astrocytes (e.g. Figure 1-amplitude increases from about 2 (spontaneous) to about 3.5 (+ATP) for WT and from about 3 (spontaneous) to about 4.5 (evoked) in MT). Therefore, the conclusion that evoked responses are different is not accurate and should be clarified.

If the increase in spontaneous and evoked calcium elevations share the same mechanism, it is rational that the scale in difference between WT and MT in spontaneous and evoked calcium activity is similar. We respectfully disagree with the reviewer that we should compare the amplitude of pharmacologically evoked calcium transients with spontaneous ones.

5) The figures are generally very difficult to follow and could use some reorganization. For example, the authors often go back and forth between human and mouse within the same figure without clearly identifying this within the figure. In addition, a number of figures could be either combined or included as supplemental material. For example, Figure 8, Figure 9, Figure 10 and Figure 11 could all be represented along with data from Figure 7 or included as a supplemental file.

We have rearranged the figures in our revised manuscript to make it easier to follow.

6) The images presented throughout the manuscript often lack many general cell markers that would benefit interpretation of the data. In some cases, the figure lacks any images of cultures (Figure 2). Most images do not include DAPI to label all cells and no images show staining for astrocyte-specific markers. Higher magnification images should also be added to the current low magnification images. In addition, the authors list quantification in the text but these data should be included in the figure to increase clarity.

We have included the requested information in figures in our revised manuscript.

7) The authors need to validate sufficient knockdown of genes and reduced protein in astrocytes following shRNAs. These data can be included as supplemental.

We did western blot to validate shRNA knockdown efficiency and included the data in our revised manuscript (Figure 7).

8) In Figure 13 and Figure 14, there is only quantification of mCherry labeling not loss of Mecp2 expression. The authors must quantify presence or absence of Mecp2 protein in mCherry positive and negative astrocytes and neurons. This is particularly important given that detecting Mecp2 in vivo in non-neuronal cells is notoriously difficult and typically requires amplification (Ballas et al., 2011). From the images, it is not clear that the antibody recognizes Mecp2 in uninfected astrocytes, which is important to show in order to validate loss of Mecp2 in infected astrocytes. This figure could also benefit from higher magnification images. Lastly, the authors suggest that they are creating cell-specific knockout mice. This is exaggerated as only about 27% of cells are infected.

We have carried out the quantification as suggested by the reviewer and included the data in our revised manuscript (Figure 8 —figure supplement 2). As for the low infection rate, it offers strong indication of the connection between loss of MeCP2 in astrocytes and the excessive activation of eNMDARs, because the phenotype of eNMDAR activation is detectable when ~1/4 of astrocytes lost MeCP2.

9) The manuscript emphasizes that their heightened extrasynaptic NMDA receptor activation has never before been reported in any RTT models. This is inaccurate. There is at least one recent paper reporting increased activation of eNMDARs in RTT (Lo et al., 2016). While it is appreciated that this effect has never been attributed to astrocytes, the text should be modified to include these published findings.

We have cited the recent publication in our revised manuscript.

10) The model provided in Figure 15 is not detailed enough to add any meaningful contribution to the manuscript as a whole and could be excluded.

We believe Figure 15 in our original submission (Figure 9 in our revised manuscript) is a good summary of our results, and can help orient readers, and prefer to keep it unless advised otherwise by the editors.

[Editors' note: the author responses to the re-review follow.]

Essential revisions:1) Most data supporting the TRPC4 mechanism are in vitro. The only data showing some evidence of increased TRPC4 function in Mecp2 null mice are immunostaining in Figure S10. More data demonstrating TRPC4 expression changes and function in vivo/ex vivo would strengthen the study. For example, western blots for TRPC4 from acutely isolated astrocytes (mouse model) in addition cultured astrocytes would be helpful.At minimum, please include in brief discussion emphasizing the need for future studies on the topic. This would help the reader and thoughtfully point out the limitations of the current work.

We agree with the reviewers that most of our data on TRPC4 are in vitro. However, current methods for isolating mouse primary astrocytes involve culturing the cells for extended period (~4 weeks), so that other cell types are removed during the process. In that sense, Western blot analysis from those cells is still in vitro. In order to assess the functional relevance of *Trpc4* in Rett syndrome disease progression, we have generated conditional *Trpc4* knockout mice, and are breeding them with the *Mecp2* knockout mice. We plan to use astrocyte-specific Cre driver to delete *Trpc4* in the *Mecp2* knockout mice and hope to report our findings in a future paper. As suggested by the reviewers, we have included a brief discussion emphasizing the need for future studies on the topic in the Discussion section. The exact language used is “As most of our TRPC4-related data came from in vitro experiments, future in vivo studies are needed to ascertain the functional significance of increased TRPC4 expression in RTT disease progression.”

2) The authors' explanation to reviewers regarding the large variability in N's is not sufficient. In many cases the N's vary by 100s of cells. For example, Figure 1C the N=526 cells for mutant cultures and N=811 for wild-type cultures. As one reviewer previously pointed out, this could greatly affect the statistical analyses. Further, it is unclear how the authors chose a particular N prior to doing statistics and it remains unclear in the manuscript how many independent experiments each data set represents.

We didn’t choose a particular N prior to doing statistics. We analyzed all cells with calcium activity in our experiments. The big difference in N between genotypes is caused by the biological difference between the genotypes. For the spontaneous calcium activity, because MECP2 mutant (MT) or knockout (KO) group usually have more cells showing calcium activity and have more frequent calcium events, resulting a bigger sample size N than that of the WT. Take Figure 1A-C as an example. 26 fields were randomly selected from each genotype group. Since the MT and WT cells were plated at the same density, the total number of cells imaged were about the same between the two genotypes. However, significantly more MT cells were observed to have spontaneous calcium activity than the WT cells (30% vs. 18%). When the subsequent analysis included only the cells with spontaneous calcium activity, the N for MT ended up being significantly bigger than the N for WT (408 vs. 280). This is the reason for the difference in N in all of our figures, except for those acute brain slice experiments. For the acute brain slice experiments, because the person performing the calcium imaging experiments was blind to the genotype of mice, it was impossible to make the N similar between the two genotype groups (especially when we typically used all mice in each litter, and those litters happened to have more WT mice than KO mice).

Moreover, we performed additional statistical analysis of all of existing data by randomly removing some values in the groups with bigger N to make the sample sizes the same between groups and included the results at the end of our response in a section named “Secondary Statistical Analysis”. For each of the figures with big difference in N, we performed such test three times (Random removal trials 1-3). All conclusions remained the same as before random removal of data (All data), suggesting the difference in N had no effect on our original conclusions. While it is not necessary to include such secondary analysis in our manuscript, we include it here to satisfy the reviewers.

Finally, more than three independent experiments/biological replicates (for human astrocyte data set, each independent experiment was defined as one independent differentiation; for mouse astrocyte in vitro data set, each independent experiment was defined as one isolation culture from one mouse; for mouse in situ and in vivo data set, both number of cells and number of mice were described) were included in each data set. Since many different types of data sets are presented, we chose to report the number of cells in most cases to make it easier to read, compare and interpret. The above language has been added to the Materials and methods section under the subheading of Statistics analysis.

3) While the authors make more attempt to validate their cell-specific genetic manipulations of Mecp2, these data are still not quantified sufficiently. The data in Supplementary file 3 and Figure 8—figure supplement 2 only shows quantification of% cells with mCherry, GFAP, and NeuN expression. The% of NeuN+ and GFAP+ cells +/- MECP2 immunoreactivity after AAV injection should also be included here. Given that the goal is to achieve cell-type specific manipulation, these data are important.

We originally focused on the mCherry signal in our quantification, because it was the fluorescence marker used for targeted recording. We thank the reviewers for making this good suggestion. Table 3 and Figure 8—figure supplement 2 using a new cohort of mice. In this experiment, we quantified the percentage of MeCP2 positive/negative cells in NeuN+ and GFAP+ cells, and reported the new results in Table 4. The top row of Table 4 confirmed our expectation that almost all the mCherry positive cells are negative for MeCP2 (i.e. virus-infected cells expressed the Cre recombinase and lacked MeCP2 expression). Nonetheless, there is negligible number (2%) of NeuN+ cells that are negative for MeCP2 in the hippocampus of AAV-GFAP-mCherry-Cre injected mice, and a small number (7%) of GFAP+ cells that are negative for MeCP2 in the hippocampus of AAV-hSyn-mCherry-Cre injected mice. These results have been added to the main text of the Result section. Furthermore, for more appropriate interpretation of our results, we have changed our conclusion of those experiments from “astrocyte-specific” and “neuron-specific” to “predominantly astrocyte-specific” and “predominantly neuron-specific”. The exact language used in our revised Result section is “Further double staining with either anti-GFAP and anti-MeCP2 antibodies or anti-NeuN and anti-MeCP2 antibodies on those brain sections revealed that, while almost all mCherry-positive cells were indeed negative for MeCP2 in mice receiving either AAV virus, there was a negligible number (2%) of NeuN-positive cells that were negative for MeCP2 in the hippocampus of the AAV-GFAP-mCherry-Cre injected mice, and a small number (7%) of GFAP-positive cells that were negative for MeCP2 in the hippocampus of the AAV-hSyn-mCherry-Cre injected mice. Thus, mice with predominantly astrocyte-specific deletion of *Mecp2* and predominantly neuron-specific deletion of *Mecp2* were generated.”

4) Many immunofluorescence images are still generally low quality. For example, in Figure 6—figure supplement 2, the background staining of TRPC4 is quite a bit higher in MECP2 null animals vs. WT. The non-specific background staining should be the same in these images to ensure a more accurate comparison. This and other figures would also benefit from some higher magnification zoom-ins in addition to the low magnification images. There is also immunostaining that is very difficult to visualize in several figures. One clear example is Figure 8—figure supplement 2 (MECP2 and NeuN immunoreactivity).

We have replaced the images in question with better quality and more representative ones (Figure 6—figure supplement 2 and Figure 8—figure supplement 2), and added higher magnification zoom-ins.